# Strong cloud–circulation coupling explains weak trade cumulus feedback

Raphaela Vogel[1,3 ✉], Anna Lea Albright[1], Jessica Vial[1], Geet George[2], Bjorn Stevens[2] & Sandrine Bony[1]

Shallow cumulus clouds in the trade-wind regions cool the planet by reflecting solar radiation. The response of trade cumulus clouds to climate change is a key uncertainty in climate projections[1–4]. Trade cumulus feedbacks in climate models are governed by changes in cloud fraction near cloud base[5,6], with high-climate-sensitivity models suggesting a strong decrease in cloud-base cloudiness owing to increased lower-tropospheric mixing[5–7]. Here we show that new observations from the EUREC⁴A (Elucidating the role of cloud-circulation coupling in climate) field campaign[8,9] refute this mixing-desiccation hypothesis. We find the dynamical increase of cloudiness through mixing to overwhelm the thermodynamic control through humidity. Because mesoscale motions and the entrainment rate contribute equally to variability in mixing but have opposing effects on humidity, mixing does not desiccate clouds. The magnitude, variability and coupling of mixing and cloudiness differ markedly among climate models and with the EUREC⁴A observations. Models with large trade cumulus feedbacks tend to exaggerate the dependence of cloudiness on relative humidity as opposed to mixing and also exaggerate variability in cloudiness. Our observational analyses render models with large positive feedbacks implausible and both support and explain at the process scale a weak trade cumulus feedback. Our findings thus refute an important line of evidence for a high climate sensitivity[10,11].

Earth's climate strongly depends on the abundance and behaviour of its smallest clouds. Shallow trade-wind cumulus clouds are rooted in the turbulent sub-cloud layer and form when thermals rise above the lifting condensation level[12]. They may grow only a few hundred metres high in dry environments or become positively buoyant and rise up to the trade-wind inversion, where they detrain condensate into stratiform cloud layers. Trade cumuli populate most of the subtropical oceans and cool the planet by reflecting the incoming solar radiation. Owing to their large geographical extent, small errors in predicting the way trade cumuli respond to warming can have a large effect on the global radiative budget. This explains why shallow cumuli in the trades are a main source of spread in the estimates of climate sensitivity of climate models[1–4].

Cloudiness near the base of the cumulus layer makes up two-thirds of the total cloud cover in the trades[13] and its change with warming governs the strength of the trade cumulus cloud feedback in climate models[5,6]. Reductions in cloud-base cloudiness in climate models are tightly coupled with increases in lower-tropospheric mixing owing to convective and large-scale circulations[5–7]. On the basis of this strong negative coupling between mixing and cloudiness, the hypothesis emerged that enhanced convective mixing might desiccate the lower cloud layer and reduce cloudiness in the trades[7]. This mixing-desiccation hypothesis suggests that the moisture transported by convection from the sub-cloud layer to the trade inversion is compensated by downward mixing of drier air and evaporation of clouds near cloud base.

The mechanism—which is expected to become more pronounced with warming owing to the nonlinear Clausius–Clapeyron relationship—is consistent with idealized high-resolution simulations of nonprecipitating trade cumuli[14] and with the behaviour of climate models that have a strongly positive trade cumulus feedback[5,7,15]. However, the mixing-desiccation hypothesis has never been tested with observations. Using the convective mass flux at cloud base, $M$, as a proxy for lower-tropospheric convective mixing, the hypothesis can be tested by analysing the relationship between $M$ and the mean relative humidity ($\mathcal{R}$) and cloud fraction ($C$) at cloud base in observations, with $C \propto \mathcal{R} \propto M^{\beta}$ and $\beta < 0$ suggesting the mixing-desiccation mechanism to be present in nature (Fig. 1a).

The mixing-desiccation mechanism is based on several assumptions that might not be operating in nature. $M$ is commonly defined as the product of the cloud fraction and the in-cloud vertical velocity, and its variability is mostly governed by the area coverage of active clouds[16,17], defined as saturated and buoyant updrafts that ventilate the sub-cloud layer. If variability in the in-cloud vertical velocity near cloud base is small, a positive relationship between $C$ and $M$ is expected ($\beta > 0$; Fig. 1b). This was demonstrated for nonprecipitating trade cumuli using Doppler radar data[17,18] and seems at odds with the mixing-desiccation hypothesis. Yet active clouds represent only half of the total $C$ (refs. [19,20]) and the lifetime and variability of passive clouds, such as the detritus of decaying clouds, might be more sensitive to $\mathcal{R}$ and mixing-induced drying of their environment than to $M$.

[1]LMD/IPSL, Sorbonne Université, CNRS, Paris, France. [2]Max Planck Institute for Meteorology, Hamburg, Germany. [3]Present address: Universität Hamburg, Hamburg, Germany. ✉e-mail: raphaela.vogel@uni-hamburg.de

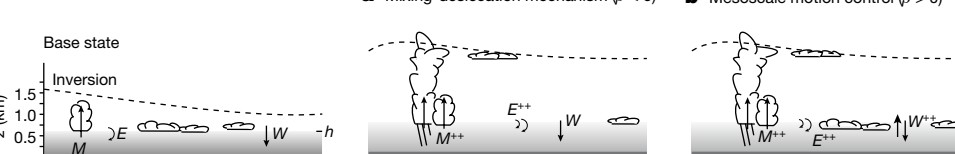

**a** Mixing–desiccation mechanism ($\beta < 0$)   **b** Mesoscale motion control ($\beta > 0$)

**Fig. 1 | Illustration of two mechanisms for the coupling of mixing and cloudiness. a**, The mixing-desiccation mechanism contends that $E$ increases in response to an increase in $M$, which leads to a reduction in $\mathcal{R}$ and cloud-base

cloudiness $C$, and a relationship $C \propto \mathcal{R} \propto M^\beta$ with $\beta < 0$. **b**, The mesoscale motion control of cloudiness instead suggests that $M$ is equally controlled by both $E$ and $W$, such that $M$ is uncorrelated to $\mathcal{R}$ and $\beta > 0$.

The sub-cloud-layer mass budget provides a theoretical basis for interpreting the mixing-desiccation mechanism. It can be expressed as a budget of the sub-cloud-layer height $h$,

$$\frac{\partial h}{\partial t} + V_h \cdot \nabla h = E + W - M, \qquad (1)$$

in which the entrainment rate, $E$, representing the mass source owing to the entrainment of dry and warm cloud layer air, and the mesoscale vertical velocity, $W$, are balanced by the mass export owing to the convective mass flux, $M$ (ref. [20]). Note that we define $M$ as the (mass) specific mass flux, which has units of velocity (see Methods). $E$ is the only term directly affecting the sub-cloud-layer moisture and heat budgets[21,22]. If an increase in $M$ is mostly balanced by an increase in $E$, a drying and warming of the sub-cloud layer and a reduction in $\mathcal{R}$ and $C$ is expected (Fig. 1a). The trades, however, exhibit strong mesoscale convective organization, which is linked to the presence of mesoscale circulations and substantial variability in $W$ (refs. [20,23–25]). This variability in $W$ could contribute to variability in $M$ without directly affecting $\mathcal{R}$ (Fig. 1b). An increase in $M$ could also produce increased inversion cloudiness and thus increased total cloud cover, compensating the radiative effects of a potential decrease in $C$. The diversity of cloud types and the large variability in $W$ in the trades thus call into question the mixing-desiccation mechanism as the dominant control of $C$ and trade cumulus feedbacks.

The EUREC⁴A field campaign was conceived to test the mixing-desiccation hypothesis[8,9]. EUREC⁴A took place in January and February 2020 near Barbados, a region selected as a source of data because clouds in its vicinity are representative for the entire trade-wind belt[26]. During EUREC⁴A, we made measurements designed to quantify the magnitude and (co-)variability of $M$, $C$ and $\mathcal{R}$ over one month, which was characterized by substantial variability in the mesoscale convective organization[27] and the large-scale circulation[9] (see Methods). With the help of these measurements, we are able to test the mixing-desiccation hypothesis with observations for the first time.

## Observations of $M$, $C$ and $\mathcal{R}$ co-variations

During EUREC⁴A, we dropped more than 800 dropsondes from the HALO aircraft flying at about 10 km altitude along 1-h circles of 220 km diameter[28,29]. We use the dropsonde data to estimate $M$ at the sub-cloud-layer top as a residual of the mass budget (equation (1)) on the 3-h scale of three consecutive circles (see Methods). Figure 2a shows a large day-to-day variability of $M$, with higher values at the beginning and end of the campaign, and a campaign mean of $17.4 \pm 7.5$ mm s⁻¹ (mean ± standard deviation $\sigma$). $M$ shows a pronounced diurnal cycle (Extended Data Fig. 1), with larger values around sunrise and smaller values in the afternoon (consistent with refs. [20,30]). The mass budget estimates are robust to changes in the estimation procedure and consistent with independent data (Methods and Extended Data Fig. 2).

The entrainment rate $E$ is computed as the ratio of the scaled surface buoyancy flux and the buoyancy jump across $h$ (equation (2) and Extended Data Fig. 3). $E$ averages to $18.3 \pm 6.4$ mm s⁻¹ across the campaign (Fig. 2b) and also shows a pronounced diurnal variability (Extended Data Fig. 1). $E$ is mostly controlled by variability in the surface buoyancy flux (Extended Data Fig. 4b). It is the strengthening of winds and surface fluxes that contributes most to the increase in $E$ and $M$ in the second half of EUREC⁴A. $W$ is, with $-0.9 \pm 6.7$ mm s⁻¹, on average nearly zero. Variability in $W$, however, is substantial and contributes slightly more to variability in $M$ compared with $E$ (Extended Data Fig. 4a). So although $M \approx E$ holds on average, consistent with the mixing-desiccation hypothesis (Fig. 1a), variability in $M$ is controlled by both $E$ and $W$.

Figure 2c shows the new measurements of the cloud-base cloud fraction $C$ from combined horizontally staring lidar and radar on board the ATR aircraft flying near cloud base[31]. $C$ is, with $5.4 \pm 3.1\%$, both small and highly variable. The variability of $C$ on the 3-h scale is substantially larger than variability on synoptic and longer timescales[13]. The robustness of $C$ is demonstrated by the internal consistency among complementary and independent measurements in terms of measurement techniques and spatial sampling[31]. The $\mathcal{R}$ at cloud base is robustly around 86% (Fig. 2d), except for a few outliers. Three data points with much lower $\mathcal{R}$ for ATR compared with HALO (marked with 'X' in Fig. 2d) are excluded in the following analyses, as these situations were associated with air masses that were sampled differently by the two aircraft (see Methods and Fig. A2 in ref. [31]).

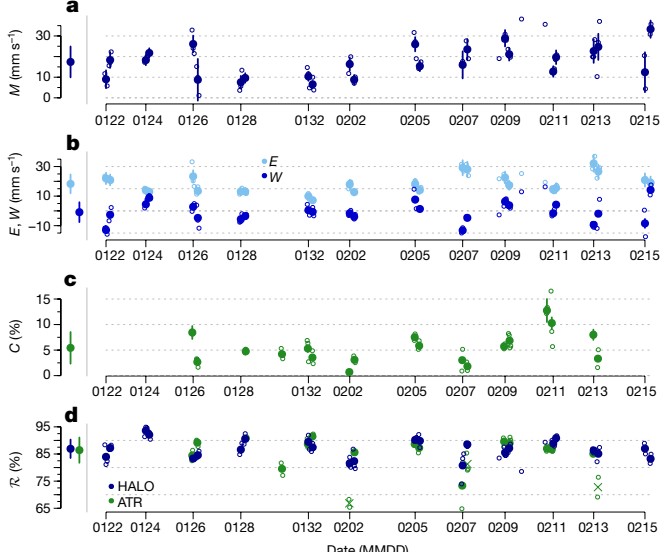

**Fig. 2 | Time series of mixing and cloudiness during EUREC⁴A. a–d**, Measurements of $M$ (**a**), $E$ and $W$ (**b**), $C$ (**c**) and $\mathcal{R}$ (**d**), with filled symbols representing the 3-h scale and open symbols representing the 1-h scale. The vertical bars in **a**–**c** show the estimation uncertainty at the 3-h scale (see Methods section 'Uncertainty estimation'). The $\mathcal{R}$ in **d** is shown for both the HALO (blue) and ATR (green) aircraft, with the 'X' markers representing the data points that are excluded in the correlations owing to inconsistent sampling of the mesoscale cloud patterns between the two aircraft. The campaign mean ± 1$\sigma$ is shown on the left side of each panel.

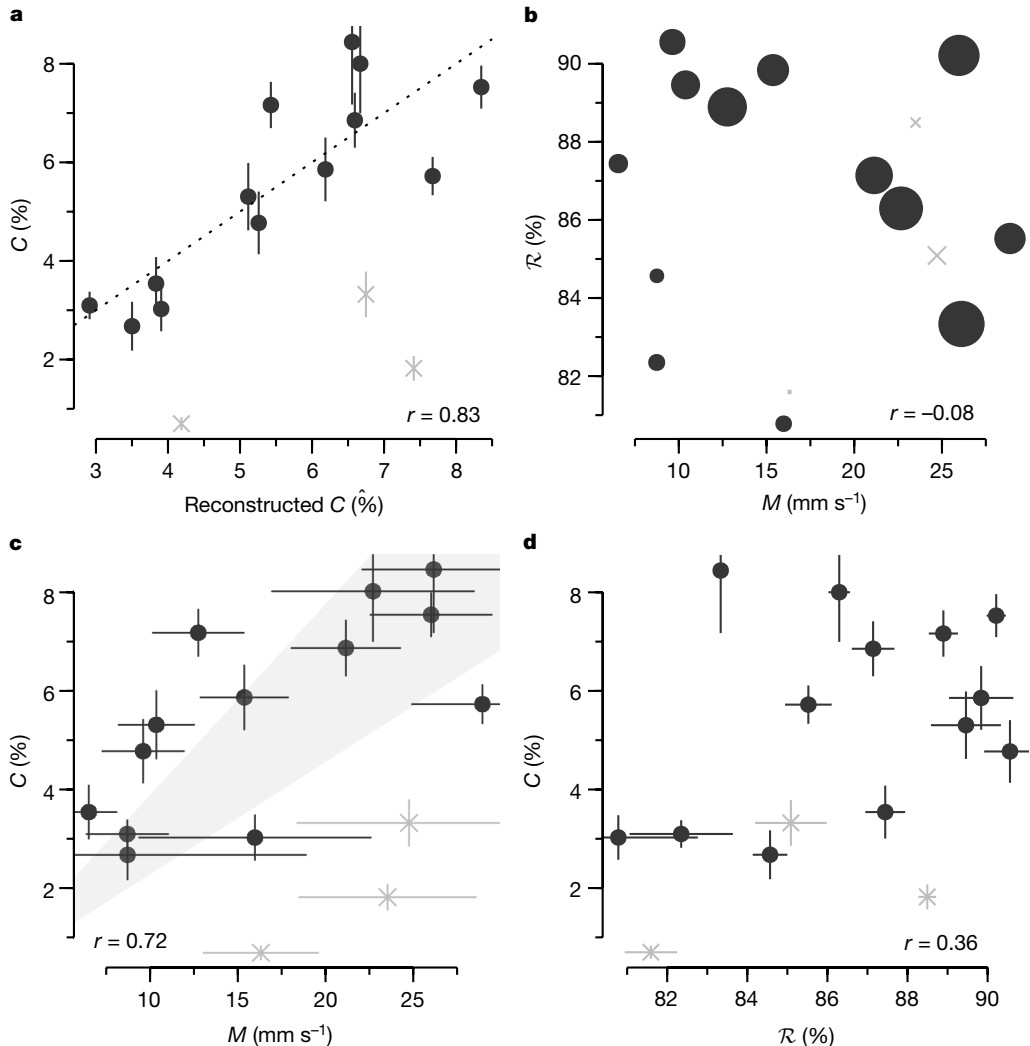

**Fig. 3 | Relationships among $M$, $\mathcal{R}$ and $C$. a–d,** The relationships between the observed $C$ and the reconstructed $\hat{C}$ from the regression $\hat{C} = a_0 + a_M\widetilde{M} + a_\mathcal{R}\widetilde{\mathcal{R}}$ (**a**), $M$ and $\mathcal{R}$ (**b**), $M$ and $C$ (**c**) and $\mathcal{R}$ and $C$ (**d**) are shown at the 3-h scale. The error bars represent the estimation uncertainty for $M$ and $C$ and the sampling uncertainty for $\mathcal{R}$ (see Methods). The dotted line in **a** is the 1:1 line. The size of the markers in **b** represents $C$. The shading in **c** represents the scaling for $C \propto 2M/w^*$ using the mean $\pm 2\sigma$ of the velocity scale $w^*$. The grey 'X' markers represent data that are excluded in the correlations owing to inconsistent sampling between the two aircraft (see Fig. 2d and Methods).

Despite being fundamental quantities to understand climate sensitivity, the challenging nature of observing $M$ and $C$ so far prevented an observational analysis of the relationship between mixing and cloud-base cloudiness. With the EUREC[4]A observations presented here, we are now able to test the mixing-desiccation hypothesis with data.

## Data refute mixing-desiccation hypothesis

The cloud-base cloud fraction is suggested to be controlled both dynamically through $M$ and thermodynamically through $\mathcal{R}$. We can therefore express $C$ as a multiple linear regression $\hat{C} = a_0 + a_M\widetilde{M} + a_\mathcal{R}\widetilde{\mathcal{R}}$, in which $\widetilde{(\ )}$ represents standardized values (for example, $\widetilde{M} = M/\sigma_M$). Figure 3a shows that the observed $C$ and the reconstructed $\hat{C}$ agree very well ($r = 0.83$ [0.80, 0.91], with values in the square brackets representing the 25th and 75th quartiles of bootstrapped correlations, respectively), demonstrating that $M$ and $\mathcal{R}$ dominate variability in $C$.

The mixing-desiccation mechanism contends that, as $M$ increases, $E$ increases and leads to a reduction in $\mathcal{R}$. The anticorrelation of $E$ and $\mathcal{R}$ is confirmed by the observations ($r_{E,\mathcal{R}} = -0.47$ [−0.62, −0.32]; Extended Data Fig. 4d). But $W$ is also correlated to $\mathcal{R}$ ($r_{W,\mathcal{R}} = 0.48$ [0.29, 0.62];

Extended Data Fig. 4e). $W$ does not directly affect the thermodynamic properties of the sub-cloud layer[22], as it transports mass with the same properties of the well-mixed sub-cloud layer. The positive correlation between $W$ and $\mathcal{R}$ is thus probably connected to a self-aggregation feedback leading to a net convergence of moisture into areas that are already moist[25,32,33]. The opposing correlations of $E$ and $W$ with $\mathcal{R}$ lead to a negligible anticorrelation of $M$ and $\mathcal{R}$ ($r = -0.08$ [−0.26, 0.10]; Fig. 3b). Although this makes $M$ and $\mathcal{R}$ independent predictors of $C$, it contrasts with the expected desiccation effect of increased mixing. The basic premise of the mixing-desiccation hypothesis thus breaks down in the presence of strong variability in $W$.

Figure 3c further shows a pronounced positive correlation between $C$ and $M$ ($r = 0.72$ [0.64, 0.81]), demonstrating that $M$ explains more than 50% of variability in $C$. The EUREC[4]A data are therefore in line with a more direct relation $C \propto M^\beta$ and a $\beta > 0$ (Fig. 1b). The tight connection between $C$ and $M$ is also consistent with physical understanding represented in the scaling $C \approx 2C_{core} \propto 2M/w^*$, in which $C_{core}$ is the area fraction of active cloud cores and $w^*$ is the Deardorff vertical velocity scale (see Methods and ref. [24]). The correlation of $C$ with $\mathcal{R}$ is weaker ($r = 0.36$ [0.16, 0.56]; Fig. 3d). These conclusions are robust to changes in the estimation procedure of $M$ and to independent estimates of $C$ (Extended Data Fig. 5).

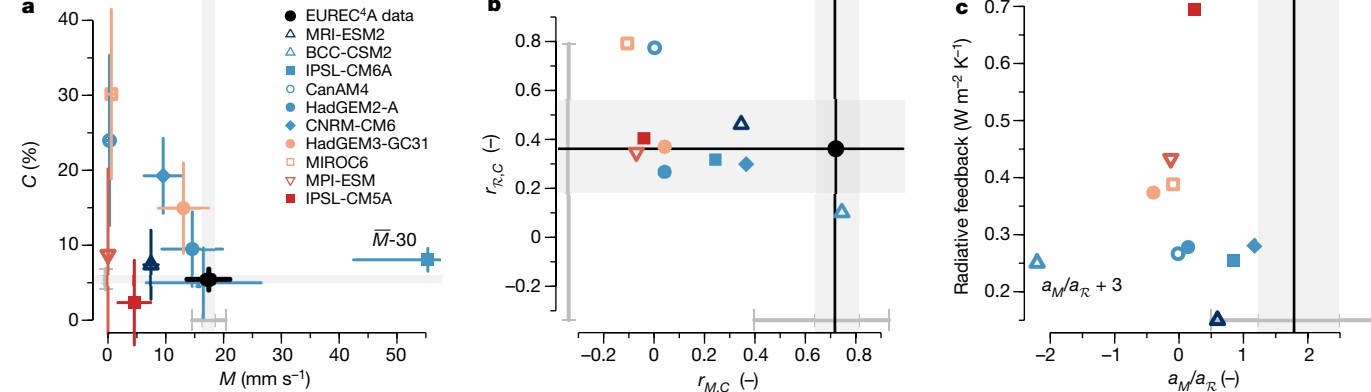

**Fig. 4 | Relationships in climate models and link to trade cumulus feedback.** **a**, Mean $\pm \sigma/2$ of $M$ and $C$. **b**, Correlation coefficients $r$ between $M$ and $C$ ($r_{M,C}$) and $\mathcal{R}$ and $C$ ($r_{\mathcal{R},C}$). **c**, Ratio of the standardized multiple linear regression coefficients $a_M/a_{\mathcal{R}}$ and the thermodynamic component of the trade cumulus radiative feedback. The models are coloured in bins of feedback strength. Open symbols refer to models with frequent stratocumulus (defined as having

$\mathcal{R} > 94\%$ more than 15% of the time; see Extended Data Fig. 8a). The grey shading represents the 25th to 75th quartile and the grey bars the 95% confidence interval of bootstrapped observational values. For plotting purposes, **a** shows the mean $\overline{M}$-30 for IPSL-CM6A and **c** shows the ratio $a_M/a_{\mathcal{R}}$ + 3 for BCC-CSM2. In **c**, the upper end of the observational 95% confidence interval (at 6.75) is cropped.

The relationships exposed by the EUREC⁴A data are thus in opposition to the mixing-desiccation hypothesis, which contends that increasing mixing (larger $M$) leads to a desiccation of the lower cloud layer (smaller $\mathcal{R}$) and a reduction in cloud-base cloudiness (smaller $C$). We also find a positive relationship between $C$ and another indicator of lower-tropospheric mixing (Extended Data Fig. 4f) and a weak positive correlation between $M$ and the total projected cloud cover (Extended Data Fig. 6). Hence, the EUREC⁴A data emphasizes dynamic factors—the convective mass flux $M$ and the mesoscale vertical velocity $W$—as dominant controls of $C$, rather than thermodynamic factors related to the mixing-desiccation mechanism.

## Models underestimate strong cloud–circulation coupling

How consistent is the present generation of climate models with our observations? To assess how climate models represent the relationship between mixing and cloudiness, we use ten models from the Cloud Feedback Model Intercomparison Project (CFMIP)[34] that provide the necessary pointwise $M$, $C$ and $\mathcal{R}$ output at high temporal resolution near the EUREC⁴A domain (see Methods). In contrast to the consistency among many independent EUREC⁴A observations, Fig. 4a shows that the models strongly differ in their magnitude and variability of $M$ and $C$. Although some models predict unrealistically low $M$ (CanAM4, MIROC6 and MPI-ESM), the IPSL-CM6A has a five times larger mean $M$ compared with the EUREC⁴A observations. Except for IPSL-CM6A, all models strongly overestimate variability in $C$ (see also ref. [35]) and 8 of 10 models also overestimate the magnitude of $C$. This is partly owing to the tendency of models to produce stratocumulus clouds in this shallow cumulus regime[36,37] (evident in the strong increases in $C$ (up to 50–100%) above a critical $\mathcal{R}$ of about 94%; see Extended Data Fig. 7). By contrast, the observations indicate no occurrence of $C > 13\%$ or $\mathcal{R} > 94\%$. The models that produce such more stratocumulus-like conditions with $\mathcal{R} > 94\%$ more than 15% of the time (Extended Data Fig. 8a) are labelled with open symbols in Fig. 4.

Only the BCC-CSM2 model represents the pronounced positive correlation between $C$ and $M$ observed during EUREC⁴A at the 3-h scale (Fig. 4b). Six of the other models have a correlation coefficient $r < 0.05$, of which three models even show a negative correlation. Most models thus strongly underestimate the tight coupling between clouds and convection observed in EUREC⁴A. Instead, these six models are more in line with the mixing-desiccation mechanism and a $\beta < 0$ (Fig. 1a), even

though this is not mediated by a pronounced negative correlation between $M$ and $\mathcal{R}$ (Extended Data Fig. 8c). All the models also strongly underestimate variability in $W$ (Extended Data Fig. 8b), as they do not represent the sub-grid processes leading to the observed variability in the mesoscale vertical velocity (for example, shallow circulations driven by differential radiative cooling[38] or local sea-surface temperature (SST) gradients[39]). The relationships between $C$ and $\mathcal{R}$ are more consistent among most models (Fig. 4b) and are also more consistent with the observations compared with the relationships between $C$ and $M$.

In contrast with the observations, clouds as parameterized by climate models are more thermodynamically than dynamically controlled. The misrepresentation of the relative sensitivity of $C$ to changes in $M$ or $\mathcal{R}$ by all models is encapsulated in the ratio of the standardized regression coefficients $a_M/a_{\mathcal{R}}$ from the regression $\hat{C} = a_0 + a_M\widetilde{M} + a_{\mathcal{R}}\widetilde{\mathcal{R}}$. The model samples lie completely outside the EUREC⁴A data (Fig. 4c). All models, with one exception, substantially underestimate the value of $a_M/a_{\mathcal{R}}$ compared with the observations, highlighting that, in the climate models, variability in $C$ is primarily controlled by variations in $\mathcal{R}$ rather than variations in $M$. Although BCC-CSM2 seems credible in terms of the magnitude and relationship of $C$ and $M$, its credibility is eroded by its unrealistic relationship between $C$ and $\mathcal{R}$ (Extended Data Fig. 7), and thus an implausible $a_M/a_{\mathcal{R}}$ of −5.2. At odds with the observations, in most models, $M$ and $\mathcal{R}$ are only weak predictors of $C$, as evident in the low coefficient of determination ($r^2$) of the multiple linear regression of $\hat{C}$ (Extended Data Fig. 8c). The cloud parameterizations of the models thus fail in capturing the key relationships between $C$ and the dynamic and thermodynamic environment observed in nature.

## Implications for trade cumulus feedbacks

The EUREC⁴A observations provide robust estimates of the mean, variability and coupling of $M$, $C$ and $\mathcal{R}$ in contrasted trade cumulus environments. Although the observed variability is substantial, the variability simulated by climate models is unrealistic, as are the drivers of this variability. The EUREC⁴A data thus provide a physical test of the capacity of models to represent the interplay of the processes active in regulating trade-wind cloud amount and may guide future model development. Moreover, the fact that the relationships at the 3-h process scale are consistent with the relationships at the monthly timescale ($r \geq 0.84$; Extended Data Fig. 8e,f) suggests that the underlying fast physical processes that couple $M$, $\mathcal{R}$ and $C$ in the models are largely

invariant with the timescale. The relationships derived from the EUREC⁴A observations can therefore also be used to evaluate the credible range of trade cumulus feedbacks in the climate models.

Figure 4b demonstrates that all models with a strong trade cumulus feedback represented by a change in the cloud radiative effect ($\Delta$CRE) with warming exceeding 0.37 W m$^{-2}$ K$^{-1}$ (reddish colours in Fig. 4c) represent the refuted mixing-desiccation mechanism with a negative (or very weak) correlation between $M$ and $C$. Also, these four models exaggerate both the coupling of $C$ to $\mathcal{R}$ (small $a_M/a_\mathcal{R}$; Fig. 4c) and the variability in $C$ ($\sigma_C$; Extended Data Fig. 8d). By contrast, the models that are closer to the observations tend to have a weaker positive $\Delta$CRE with warming. The EUREC⁴A observations of the physical processes that drive the short-term variability of $C$ thus rule out the mechanism that leads to the largest positive trade cumulus feedbacks in current climate models.

By showing that mesoscale motions inhibit the mixing-desiccation mechanism, we refute an important physical hypothesis for a large trade cumulus feedback. In the spirit of the storyline approach for constraining equilibrium climate sensitivity[10], our findings thus refute an important line of evidence for a strong positive cloud feedback and thus a large climate sensitivity. The EUREC⁴A observations therefore support recent satellite-derived constraints from observed natural variability[37,40] and climate-change experiments using idealized high-resolution simulations[41,42], which suggest that a weak trade cumulus feedback is more plausible than a strong one. Moreover, for the first time, we take into account all types of cloud present in the trades, including the optically thinnest ones that are usually missed in satellite observations[43], and consider the full range of mesoscale variability that was not represented in idealized simulations of cloud feedbacks. We also provide an explanation for the inconsistency of models with large positive feedbacks: in these models, the observed tight coupling between convective mixing and cloudiness is absent; instead, $C$ is primarily controlled thermodynamically by $\mathcal{R}$, which exaggerates variability in $C$ and feedbacks to warming. By not representing the variability in mesoscale circulations, the models miss an important process regulating trade cumulus clouds. Future research should focus on better understanding the processes controlling these mesoscale circulations and how they might change in a warmer climate.

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

**Publisher's note** Springer Nature remai utional affiliations.

# Methods

## EUREC⁴A field campaign

We use data from the EUREC⁴A field campaign, which took place in January and February 2020 and was anchored in Barbados[8,9]. We focus on measurements made by the HALO[29] and ATR aircraft[31], which flew coordinated patterns in the approximately 220-km diameter EUREC⁴A circle centred at 13.3° N, 57.7° W. The HALO aircraft flew three circles at 10.2 km altitude in 200 min (about 60 min per circle plus a 15-min break between circles) and launched dropsondes every 30° of heading (about 12 sondes per circle) to characterize the large-scale environment[28]. At the same time, the ATR aircraft flew 2–3 50-min rectangle patterns inside the circle near cloud base and measured the cloud fraction with horizontally staring cloud radar and backscatter lidar, and with several in situ probes and sensors[31]. Observations from the Barbados Cloud Observatory (BCO)[15] and the RV Meteor[44] provide further context at the western and eastern boundaries of the EUREC⁴A circle.

A typical flight day of HALO consisted of two sets of three consecutive circles lasting about 3 h and comprising 30–36 sondes (sometimes defined as circling[9,22,29]). The 3-h circle sets are separated by a 1.5-h break to refuel the ATR. The circle patterns were flown from 22 January to 15 February with different starting times between 04:30 and 19:30 local time (LT) to sample the diurnal cycle. Four more single-dropsonde circles are also used, three of which were flown by the P3 aircraft[45] during nighttime (starting at 00:15 LT on 9 and 10 February, and at 01:30 LT on 11 February). In total, the dataset comprises 73 circles (1-h scale) and 24 sets of three consecutive circles (3-h scale), for which 16 have coincident ATR data. We assume that HALO and ATR sample comparable conditions on the 3-h scale. This is confirmed by the similar cloud-base $\mathcal{R}$ of the aircraft during most flights (Fig. 2d), except for the first 3-h circle set on 2 February and the second 3-h circle set on 7 and 13 February, in which the spatial scale of the cloud organization was larger than the scale of the domain sampled by the ATR. These three 3-h circle sets are marked in the figures and excluded from the calculated correlations.

The spatial scale of the observations represents the lower end of Orlanski's[46] meso-$\alpha$ scale and is comparable in size with a climate model grid box. The 200–300-km scale is the relevant scale of the cloud processes for a trade cumulus ensemble and also the scale that convective parameterizations target. It lies in between the $\mathcal{O}$(1 km) scale of individual clouds and the synoptic scale of $\mathcal{O}$(1,000 km), and is associated with the emergence of the prominent trade cumulus cloud organization patterns[47]. As the air masses are advected by about 30 km per hour (at the campaign mean wind speed of about 9 m s⁻¹ at 1 km height), the spatial sampling of the 220-km diameter circle does not differ substantially between the 1-h and 3-h timescales, which motivates our nomenclature focus on the time rather than space scale. Using the measurements, model and reanalysis data, we would not expect our results to change substantially if the analysis domain were increased or reduced by a factor of two or more (see Methods section 'Mass flux estimation' for a discussion of the scale sensitivity of the results).

The Barbados region was chosen as the location of EUREC⁴A because shallow trade cumulus clouds are the dominant cloud type in the area during winter[13]. Furthermore, clouds in the Barbados region are similar to clouds across the trade-wind regions in both observations and models[26]. The mean meteorological conditions during the EUREC⁴A campaign, as sampled by the dropsondes, also correspond well to the average January–February conditions from 12 years of data from the ERA-Interim reanalysis[48] (their Fig. 5), albeit with a 10% larger 850 hPa relative humidity during EUREC⁴A (the EUREC⁴A dropsondes also have an approximately 8% larger relative humidity compared with the 2013–2022 average in ERA5, not shown). Also, all four prominent patterns of mesoscale cloud organization[47] were present during the campaign[27]. The conclusions drawn from the EUREC⁴A data are thus relevant across the tropics and for climate timescales.

## Observations

For estimating the cloud-base mass flux $M$, $\mathcal{R}$ and many other variables, we use dropsonde data from the JOANNE dataset[28], namely Level 3 (gridded quality-checked sondes) and Level 4 (circle products) vertical profiles of thermodynamic quantities, wind and mesoscale vertical velocity, $W$. The HALO dropsondes are corrected for a dry bias by multiplying the relative humidity by 1.06 (ref. [28]).

For the cloud-base cloud fraction $C$, we use the BASTALIAS lidar-radar synergy product[31], which includes both cloud and drizzle (but not rain) and constitutes an upper bound on $C$. We also test the relationships for three further estimates of $C$:
- The non-drizzling cloud product from the radar-lidar synergy ($C_{\mathrm{only}}$), which excludes drizzle and constitutes a lower bound on $C$.
- In situ estimates from a microphysical probe defined on the basis of thresholds of liquid water content plus particle size ($C_{\mathrm{pma}}$).
- In situ high-frequency (25-Hz) humidity sensor, with cloud defined as relative humidity ≥98% ($C_{\mathrm{turb}}$).

The in situ sensors measure the along-track $C$, whereas the lidar-radar synergy samples clouds inside the rectangle at a distance up to 8 km from the aircraft[31]. Despite pronounced differences in the measurement principles and sampling, Fig. 18 of ref. [31] demonstrates the internal consistency and robustness among the independent $C$ estimates. The ATR turbulence measurements also include measurements of vertical updraft and downdraft velocities[49], from which an in-cloud mass flux $M_{\mathrm{turb}}$ is computed by multiplying $C_{\mathrm{turb}}$ by the in-cloud vertical velocity.

Further HALO aircraft measurements used are total projected cloud cover (CC) estimates from the differential absorption lidar WALES, the hyperspectral imager specMACS and the cloud radar HAMP[29]. From these cloud masks, we derive the CC along the 1-h circle. For specMACS and HAMP, the cloud detection is ambiguous and we consider both the 'probably cloudy' and the 'most likely cloudy' flags in our CC estimates.

We also use ceilometer and cloud radar data from the BCO and the RV Meteor to test the robustness of the sub-cloud-layer height definition (not shown). Radar cloud fraction profiles are obtained by correcting the hydrometeor fraction profiles with ceilometer data during periods of rain (see ref. [30] for a description of the correction applied). The BCO cloud radar data also demonstrate that missing the level of maximum cloud-base cloud fraction in 3-h averages by, say, 60 m does not affect the variability of $C$ (correlations of $r = 0.99$ and $r = 0.93$ with the maximum $C$ when 60 m above and below the peak level, respectively) and only marginally affects its magnitude (18% and 33% smaller relative to the maximum $C$ for being 60 m above or below the peak level, respectively). So only if the ATR flight level deviated from the height of maximum cloudiness in ways that co-varied with $M$ would we expect such a height difference to influence our analysis. As the ATR aircraft usually flew slightly above $h$ (Extended Data Fig. 3a) and because it sampled many more clouds in 3 h compared with the stationary BCO, a potential influence of missing the peak level is deemed not to bias our findings.

## Surface buoyancy flux

To estimate the surface buoyancy flux ($\overline{w'\theta'_{\mathrm{v}}}|_{\mathrm{s}}$, needed to compute $M$), we use dropsonde humidity, temperature and wind data at 20 m height and apply the Coupled Ocean-Atmosphere Response Experiment (COARE) bulk flux algorithm version 3.6 (refs. [50,51]). For the SST, we extrapolate the 2-m-depth SST of the RV Meteor (thermosalinograph primary backboard temperature), or alternatively from the AutoNaut Caravela[52], to the dropsonde location based on a fixed zonal and meridional SST gradient of −0.14 K per degree. A gradient of −0.14 K per degree corresponds to the median zonal and meridional gradient (−0.145 K per degree and −0.135 K per degree, respectively) across the EUREC⁴A circle over the period from 19 January to 15 February in the ERA5 reanalysis[53] and in two satellite SST products (from the Advanced Baseline Imager on board the Geostationary Operational Environmental Satellite (GOES-16 ABI) and the Collecte Localisation Satellites (CLS).

The sonde-derived surface buoyancy flux on the 3-h scale compares favourably with bulk fluxes from the RV Meteor mast, with a correlation coefficient $r = 0.83$ and a mean offset of 0.1% relative to RV Meteor. The sonde-derived flux has a comparable magnitude with the flux measured at the RV Ronald Brown[54] further upstream and is also well correlated ($r = 0.81$) with ERA5. The ERA5 fluxes, however, overestimate the surface buoyancy flux compared with the sonde-derived flux by 25%, which is mostly because of the overestimation of the sensible heat flux by 64% relative to the observations (9.8 W m$^{-2}$ and 6.0 W m$^{-2}$ for ERA5 and dropsondes, respectively). A strong overestimation of the sensible heat flux compared with buoy measurements in the region is also present in the predecessor ERA-Interim reanalysis[55]. Overall, the good correspondence of our sonde-derived surface buoyancy flux with the independent data lends credibility to our estimation procedure. The sonde-derived surface buoyancy flux is also used to compute the Deardorff sub-cloud-layer vertical velocity scale $w^* = \left(h \frac{g}{\theta_v} \overline{w'\theta_v'}\big|_s\right)^{1/3}$ shown in Fig. 3c, in which g is the gravitational acceleration.

## Mass flux estimation

Vogel et al.[20] developed a method to estimate the shallow-convective mass flux at the sub-cloud-layer top as a residual of the sub-cloud-layer mass budget and tested it in real-case large-eddy simulations over the tropical Atlantic. Here the method is applied to EUREC⁴A observations, in parallel with Albright et al.[22], who close the sub-cloud-layer moisture and heat budgets and provide an independent constraint on the entrainment rate $E$. Except for the surface buoyancy flux estimate (see the previous section), all data for the budgets come entirely from the dropsondes.

Equation (1) expresses the budget of the sub-cloud-layer height $h$ per unit area and constant density. $\frac{\partial h}{\partial t}$ represents the temporal fluctuation of $h$ and $V_h \cdot \nabla h$ its horizontal advection, $E$ is the entrainment rate, $W$ the mesoscale vertical velocity (positive upwards) and $M$ the convective mass flux at $h$.

The sub-cloud-layer height $h$ is defined as the height at which the virtual potential temperature ($\theta_v$) first exceeds its density-weighted mean from 100 m up to $h$ by a fixed threshold $\epsilon = 0.2$ K (refs. [22,56]). Extended Data Fig. 3a confirms that our $h$ is usually close to the ATR flight altitude and $h$ is also well within the range of independent BCO and RV Meteor observations of the maximum radar cloud-base cloud fraction and the peak frequency of the first ceilometer cloud-base height (not shown). This confirms that our $h$ agrees well with the level of maximum near-base cloud fraction, which was set as the target height for the ATR flight level and thus for evaluating the mass budget[31].

The entrainment rate $E$ represents the deepening of $h$ owing to small-scale mixing at the sub-cloud-layer top. We use a modified version of the classical flux-jump model[57,58] that accounts for the finite thickness of the transition layer, the approximately 150-m-thick stable layer separating the mixed layer from the cloud layer (see ref. [22] for details). The buoyancy flux at $h$ is modelled as a fixed fraction $A_e$ of the surface buoyancy flux, $\overline{w'\theta_v'}\big|_s$, in which $A_e$ is the effective entrainment efficiency. The buoyancy jump at the sub-cloud-layer top is computed as $\Delta\theta_v = \Delta\theta + 0.61(\overline{\theta}\Delta q + \overline{q}\Delta\theta)$, with $\Delta\theta = C_\theta(\theta_{h+} - \overline{\theta})$ and $\Delta q = C_q(q_{h+} - \overline{q})$. $q$ is the specific humidity, $C_q$ and $C_\theta$ are scaling coefficients accounting for uncertainty in the depth over which the jumps are computed, the subscript $h+$ refers to the value of $q$ or $\theta$ above $h$ (computed as the average from $h$ to $h + 100$ m) and $\overline{q}$ and $\overline{\theta}$ are averages from 50 m to the mixed-layer top (defined as the height of maximum relative humidity below 900 m). Finally, $E$ is computed as

$$E = \frac{A_e \overline{w'\theta_v'}\big|_s}{\Delta\theta_v} \tag{2}$$

The uncertain parameters $A_e$, $C_q$ and $C_\theta$ are estimated through a joint Bayesian inversion to close the moisture and heat budgets by ref. [22],

yielding maximum-likelihood estimates of $A_e = 0.43 \pm 0.06$ (mean $\pm 1\sigma$), $C_q = 1.26 \pm 0.34$ and $C_\theta = 1.15 \pm 0.31$.

The mesoscale vertical velocity $W$ at $h$ is computed by vertically integrating the divergence of the horizontal wind field measured by the dropsondes[23] from the surface up to $h$. $W$ is at the lower end of the meso-$\alpha$ scale of ref. [46], what climate modellers often associate with the 'large scale'. The terms $h$, $E$ and $W$ are computed at the 1-h scale of a single circle and then aggregated to the 3-h scale (three circles).

The temporal fluctuation of $h$ is estimated as the linear regression slope of $h$ computed from the 30–36 soundings available per 3-h circle set. Similarly, the horizontal advection of $h$ is estimated as the sum of the linear regressions of the eastward ($\partial h/\partial x$) and northward ($\partial h/\partial y$) gradients of the individual $h$, multiplied by the wind speed at the 3-h mean $h$. Both $\partial h/\partial t$ and $V_h \cdot \nabla h$ are only available on the 3-h scale.

The default $M$ shown in the paper is the equilibrium mass flux $M = E + W$, which reproduces well the mass flux diagnosed directly from cloud-core area fraction and vertical velocity in large-eddy simulations[20]. This equilibrium $M$ is also available on the 1-h scale of an individual circle. Taking into account $\partial h/\partial t$ and $V_h \cdot \nabla h$ in the mass flux estimate leads to $M' = M - \frac{\partial h}{\partial t} - V_h \cdot \nabla h$, which shows very similar characteristics compared with $M$ (Extended Data Fig. 3). This is mainly because both the advection ($-1.3 \pm 2.7$ mm s$^{-1}$) and temporal fluctuation ($0.5 \pm 6.8$ mm s$^{-1}$) terms are on average about zero, and the advection term is also nearly invariant. The inclusion of advection and $\frac{\partial h}{\partial t}$ in $M'$ slightly enhances variability on the diurnal timescale (Extended Data Fig. 1a).

Cold pools formed by evaporating precipitation destroy the structure of the sub-cloud layer and make the estimation of $h$ less robust. We thus exclude soundings that fall into cold pools in the analysis using the criterion of $h < 400$ m developed by ref. [56] based on the EUREC⁴A soundings. The influence of these and other assumptions on the magnitude and variability of $M$ are discussed in the Methods section 'Robustness of observational estimates'. Also note that our $M$ is defined as the (mass) specific mass flux and has units of velocity. It differs from the more familiar mass flux (in units of kg m$^{-2}$ s$^{-1}$) by the air density, which is usually assumed to be constant[18,59], and which is justified here given the small density variations across the measurements (mean $\pm \sigma$ of $1.104 \pm 0.0077$ kg m$^{-3}$, that is, less than 0.7% of the mean).

Although the 1-h scale variability of $M$ can be substantial (for example, second 3-h circle sets on 26 January and 13 February; Fig. 2), the median estimation uncertainty is only 20% at the 3-h scale (see section below). Also, $M$ has a similar magnitude and reassuring correlation ($r = 0.77$) to the independent $M_{turb}$ estimate from in situ turbulence measurements on the ATR aircraft (Extended Data Fig. 2d).

The mass budget terms show different degrees of scale sensitivity (see also discussion in ref. [20]). Extended Data Figs. 2c and 4a show that the correlation between $W$ and $M$ is slightly larger at the 1-h scale compared with the 3-h scale ($r_{W,M3h} = 0.60$ and $r_{W,M1h} = 0.67$), whereas they are essentially the same for $E$ and $M$ ($r_{E,M3h} = 0.54$ and $r_{E,M1h} = 0.55$). The scale sensitivity of the $W$ variance is in line with radiosonde data from the EUREC⁴A ship array, which show that the divergence amplitudes at equivalent radii of 100–300 km scale inversely with radius[60] (as in ERA5 and ICON, consistent with ref. [23]). In ERA5, the scale sensitivity of the surface buoyancy flux, which contributes most to variability in $E$ (Extended Data Fig. 4b), is much smaller compared with the scale sensitivity of $W$ (not shown). This is probably because variability in the surface buoyancy flux is mostly controlled by the surface wind speed (Extended Data Fig. 4h) and radiative cooling[61], both of which are large scale. The surface wind speed has autocorrelation coefficients of 0.74 for a 2-day and 0.48 for an 8-day lag (Fig. 3d of ref. [22]). Although weaker compared with the synoptic variability, the surface wind also has a distinct diurnal cycle[62,63], which causes a diurnal cycle of the surface buoyancy flux (Extended Data Fig. 1c and ref. [20]). Some of the diurnal variability in $E$ is thus lost for longer temporal averaging. Also, the variability in the temporal fluctuation and horizontal advection of $h$ (equation (1)) decreases on larger scales[20]. In summary, $M$ variability decreases on

larger averaging scales. The scale sensitivity of $W$ is larger compared with $E$, such that the contribution of $W$ to $M$ variability tends to become smaller compared with the contribution of $E$ on much larger scales.

As noted above, $E$ describes the net effect of local processes and must be inferred from the statistics of other quantities (that is, the mean sub-cloud-layer growth rate or the dilution of sub-cloud-layer properties). This raises the question whether the $E$ estimate itself might depend on the mesoscale environment and therefore introduce spurious co-variabilities between $M$, $W$ and $C$. The Bayesian estimation of the uncertain parameter estimates $A_e$, $C_q$ and $C_\theta$ is a priori independent of $M$ and $W$. Also, the synoptic variability during EUREC[4]A can be well explained by keeping them constant[22]. Reference [22] also explored to what extent other factors correlated with residuals in their Bayesian fits and found no evidence of a systematic effect of other factors, including wind speed and shear[64]. As discussed above, the variability in $E$ tends to be less scale-sensitive than $W$ and mostly controlled by larger-scale factors, such as the surface wind speed (through the surface buoyancy flux; Extended Data Fig. 4b,h). Furthermore, $E$ and $W$ are anticorrelated ($r_{E,W} = -0.35$; Extended Data Fig. 4g). So both statistically from the anticorrelation and physically through the scale argument, we believe that our parameterization of $E$ does not induce spurious co-variability.

## Uncertainty estimation

For the $M$, $\mathcal{R}$ and $C$ estimates, we distinguish two sources of uncertainty: sampling uncertainty and estimation (or retrieval) uncertainty. For all terms, the sampling uncertainty is computed at the 3-h scale as the standard error, $SE = \sigma/\sqrt{n}$, of the three individual 1-h circle values (each representing about 50 min of flight or up to 12 sondes), in which $\sigma$ is the standard deviation and $n$ the number of circles.

The estimation uncertainty is computed differently for every term according to the underlying assumptions and choices. For $\mathcal{R}$, the manufacturer-stated uncertainty (that is, repeatability) is 2% and some extra uncertainty stems from the correction of the dry bias of the HALO dropsondes (see ref. [28]). Because this uncertainty is the same for all data points, the estimation uncertainty of $\mathcal{R}$ is not shown in the figures. For $C$, the estimation uncertainty is computed for every 3-h circle set as the SE of the four different estimates of $C$, namely $C$ itself, $C_{only}$, $C_{turb}$ and $C_{pma}$. The uncertainty estimate therefore represents uncertainty in measurement principles and spatial sampling[31]. Further uncertainties of the individual $C$ estimates (for example, owing to the choice of thresholds) are neglected, as sensitivity tests suggest that they are smaller than the uncertainty among the different $C$ estimates[31].

For $W$, the advection term $V_h \cdot \nabla h$ and the temporal fluctuation $\partial h/\partial t$, the estimation uncertainty is taken as the SE of the respective regression used to compute the term. Because $\partial h/\partial t$ is computed from individual sondes, it contains both temporal and spatial variability of $h$ on the 3-h scale and its SE is inflated.

The estimation uncertainty of the surface buoyancy flux is a combination of uncertainty in the underlying SSTs and in the COARE bulk flux algorithm. We estimate the uncertainty in the underlying SSTs by computing the SE of five different versions of the flux (three with different fixed SST gradients (the default median value and the median ± interquartile range, that is, −0.14 K per degree, −0.21 K per degree and −0.07 K per degree), one with a temporally varying gradient (not shown) and one with a different baseline SST (from the AutoNaut Caravela[52] instead of the RV Meteor)) and adding a 5% uncertainty of the COARE algorithm given in ref. [50] as the 1-h uncertainty in the 0–10 m s⁻¹ wind speed range. For $A_e$ and $\Delta\theta_v$, we use the relative uncertainties of the Bayesian inversion as the estimation uncertainty (that is, $\sigma(A_e)/A_e$ for $A_e$ and the average of $\sigma(C_q)/C_q$ and $\sigma(C_\theta)/C_\theta$ for $\Delta\theta_v$).

Uncertainties in the three individual terms of $E$ are propagated by adding their fractional uncertainties in quadrature to yield the estimation uncertainty of $E$. In the same spirit, the estimation uncertainty is propagated from the 1-h scale to the 3-h scale and from the individual terms of equation (1) to $M$.

The uncertainties of the correlations and the multiple linear regression are estimated with bootstrapping (10,000 repetitions). We communicate these uncertainties by mentioning the 25th and 75th quartiles in the text and by showing both the quartiles and the 2.5% and 97.5% quantiles (representing the 95% confidence interval) in Fig. 4 and Extended Data Fig. 8. Apart from the uncertainty quantification described here, we assess the robustness of the $M$ and $C$ observations to several other choices and assumptions in the Methods section 'Robustness of observational estimates'.

## Other mixing indicators

Other proxies for lower-tropospheric mixing were used in previous studies[5,7,65] that can be estimated from the dropsonde data and compared with the variability in $C$. Here we compute the boundary-layer vertical advection (BVA) diagnostic from ref. [65] defined as $BVA = \int_0^{Z_{min}} W(z)\frac{\partial MSE}{\partial z}\rho dz$, in which MSE is the moist static energy, $Z_{min}$ the level of minimum MSE that marks the top of the trade-wind layer (on average at 2,900 m) and $\rho$ the density. Note that a lower (more negative) BVA value indicates stronger mixing.

Reference [65] found a pronounced positive relationship between changes in BVA and changes in $C$ from a series of single-column model experiments with the IPSL-CM5A model, which is characterized by a strong positive low-cloud feedback and the presence of the mixing-desiccation mechanism (Fig. 4). Extended Data Fig. 4f shows a pronounced negative correlation between BVA and $M$ in the EUREC[4]A data, indicating good agreement in their complementary definitions of mixing. Smaller BVA (stronger mixing) is also associated with larger $C$ (not shown), which is at odds with the IPSL-CM5A model. The absolute correlation between BVA and $C$ ($r = 0.34$), however, is considerably smaller than the correlation between $M$ and $C$ ($r = 0.72$).

## General circulation models

The cloud fraction, net mass flux (upward and downward) and relative humidity at cloud base are calculated for ten Coupled Model Intercomparison Project (CMIP) models:
- Four from the fifth phase, CMIP5 (ref. [66]): CanAM4 (ref. [67]), MPI-ESM-LR[68], IPSL-CM5A-LR[69], HadGEM2-A[70] and
- Six from the sixth phase, CMIP6 (ref. [71]): BCC-CSM2-MR[72], CNRM-CM6-1 (ref. [73]), IPSL-CM6A-LR[74], MIROC6 (ref. [75]), MRI-ESM2-0 (ref. [76]), HadGEM3-GC31-LL[77]

using the sub-hourly vertical profiles at selected sites (named cfSites in CMIP5 and $CF_{subhr}$ in CMIP6) provided by CFMIP[34]. Note that the $M$ from the models is not computed using equation (1) but is defined according to the respective convective parameterization scheme of the models (see references above). We use the atmosphere-only amip configuration from 1979 to 2008, selecting data from December, January, February and March to be broadly consistent with the winter conditions sampled during EUREC[4]A. For each model, between two and six sites are available in the North Atlantic trades between 60–50° W and 12–16° N, namely the BOMEX, NTAS, EUREC[4]A, BCO and RICO sites. All profiles with clouds above 600 hPa (about 4.2 km) are dropped to ensure a focus on shallow convection. We verified that, in terms of the large-scale environment, the cfSites fall into the climatological trade cumulus regime as defined by ref. [40].

The cloud-base level is defined as the level of maximum cloud fraction between 870 and 970 hPa (between about 400 and 1,300 m). If the maximum cloud fraction is smaller than 0.25% for a given profile, the cloud-base level is taken at the climatological level of maximum cloud fraction. The hourly cloud-base data are aggregated to a 3-h timescale, which corresponds to the 3-h scale of the EUREC[4]A data, as well as a monthly timescale. The values computed are insensitive to (1) averaging across the sites before aggregating to the 3-h timescale, (2) removing the site near the Northwest Tropical Atlantic Station buoy upstream of the EUREC[4]A circle (near 51° W and 15° N), (3) focusing only

on January and February, and (4) excluding nighttime values outside the hours sampled during EUREC[4]A (not shown).

We use the thermodynamic component of the change in the cloud radiative effect at the top of the atmosphere ($\Delta$CRE) with warming under given dynamical conditions to quantify the strength of the trade cumulus radiative feedback. Reference [2] showed that the $\Delta$CRE with warming is a good approximation of the cloud feedback computed with radiative kernels[78]. The CRE is defined as the difference between all-sky (all, including clouds) and clear-sky (clr, clouds assumed to be transparent to radiation) net downward radiative fluxes, $\text{CRE} = R_{all} - R_{clr} = (LW_{clr} - LW_{all}) + (SW_{all} - SW_{clr}) = \text{CRE}_{LW} + \text{CRE}_{SW}$, with $R$ being the total radiative flux and LW and SW its longwave and shortwave components, respectively. The radiative fluxes are defined positive downward. The $\Delta$CRE with warming is then simply the difference in CRE between the warmer amip4K (4-K uniform increase in SST) and the amip (control) simulations, normalized by the 4-K temperature difference (that is, $\Delta$CRE/$\Delta T_s = (\text{CRE}_{amip4K} - \text{CRE}_{amip})/4\text{K}$). To restrict the feedback estimation to the trade cumulus regime, we focus on ocean-only grid points between 35° S and 35° N and use the regime partitioning of ref. [40] with trade cumulus regimes in each simulation (amip or amip4K) defined as having a climatological annual mean estimated inversion strength smaller than 1 K and a vertical velocity at 700 hPa between 0 and 15 hPa day$^{-1}$.

## Robustness of observational estimates

Applying the mass budget formulation to the EUREC[4]A dropsonde data involves several choices for definitions and thresholds. These choices are guided by constraints from independent data and from closure of the moisture and heat budgets in ref. [22], which provides justification for the default configuration described in the Methods section 'Mass flux estimation'. Nevertheless, it is important to assess and understand the sensitivity of the mass budget estimates and the key relationships to different estimation choices.

We focus first on the influence of different definitions of the sub-cloud-layer height $h$ and the entrainment rate $E$ on the mean and standard deviation ($\sigma$) of $M$ and $E$, the respective correlations of $M$ with $E$ and $W$, and the correlation and mean difference to the independent $M_{turb}$ estimate from turbulence measurements onboard the ATR aircraft (see Extended Data Fig. 2). For the $h$ definition, we compare our default $h$ to an alternative definition, 'h.parcel', which defines $h$ as the level of neutral buoyancy of a surface-lifted parcel (with density-weighted $\theta_v$ averaged from 30 to 80 m) plus 0.2 K $\theta_v$ excess. Using 'h.parcel' leads to a 16 m shallower mean $h$ compared with the default. The slightly shallower $h$ decreases $\Delta\theta_v$ (the denominator of the $E$ formulation in equation (2)) from 0.36 K to 0.34 K, which slightly increases $E$ and $M$ by around 1.5 mm s$^{-1}$. Although $W$ is unaffected by this small change in $h$, the resulting $M$ has a slightly reduced correlation to the independent $M_{turb}$ compared with the default $M$ ($r = 0.69$ versus $r = 0.77$). The same chain of arguments holds for increasing and decreasing the threshold $\epsilon$ in the $h$ definition by ±0.05 K. With $\epsilon = 0.25$ K instead of 0.2 K (case 'h.eps = 0.25'), $h$ increases by 31 m and, through the larger $\Delta\theta_v$, decreases $E$ and $M$ by about 3.3 mm s$^{-1}$. Owing to the presence of a thin transition layer[22], the response to $\epsilon$ ± 0.05 K is nonlinear and a reduction of $\epsilon$ to 0.15 K ('h.eps = 0.15') leads to a disproportionately smaller $\Delta\theta_v$ and roughly 6 mm s$^{-1}$ larger $E$ and $M$. The 35 m shallower mean $h$ with $\epsilon = 0.15$ K also strongly increases $\sigma_E$, which increases the correlation between $E$ and $M$ at the expense of a decreased correlation between the unaffected $W$ and $M$ (Extended Data Fig. 2c).

The next set of choices involves the entrainment rate estimate. We test the influence of the different surface buoyancy flux estimates from ERA5 and RV Meteor. As the ERA5 flux is 25% larger than the other fluxes, we scale it to have same mean as the dropsonde-derived flux (case 'sbf = ERA5.sc'). For 'sbf = ERA5.sc', the variability in $E$ and $M$ are substantially larger compared with the default dropsonde flux, increasing their correlation. For the case 'sbf = Meteor', the differences to the default estimates is smaller (Extended Data Fig. 2a,b) and the correlation with

$M_{turb}$ is slightly larger than in the other configurations. The estimates are also unaffected by changing the three coefficients $A_e$, $C_q$ and $C_\theta$ estimated by Bayesian inversion in ref. [22] to close the moisture and energy budgets during EUREC[4]A when cold pool soundings (defined as having $h < 400$ m following ref. [56]) are excluded ('diffEpars'). We further compare four different ways of computing $\Delta\theta_v$. Computing the value at $h+$ as averages from $h$ to $h + 50$ or $h + 150$ m (instead of to $h + 100$ m) has a similar (but more linear) influence as increasing $\epsilon$ ± 0.05 K (see discussion above). Using two different heights for averaging $\theta_v$ across the mixed layer (up to $h$ in 'tvbar = h' and up to the level at which $q$ first falls below its mean by a threshold of 0.3 g kg$^{-1}$ in 'tvbar = qgrad') hardly influences the estimates.

Last, we show the influence of computing the mass budget including the cold pool soundings for two sets of surface buoyancy flux estimates, case 'withCP' for the default dropsonde-derived flux and 'withCP_sbf = ERA5.sc' for the scaled ERA5 flux. In both cases, the mean and $\sigma$ of both $M$ and $E$ are increased when cold pools are included (matching the mean $E$ of ref. [22], who included cold pools). However, especially for the default surface fluxes (case 'withCP'), the correlation with $M_{turb}$ is strongly reduced.

Extended Data Fig. 2a,d also shows the influence of selected choices on the total mass flux $M'$, which includes the contribution of the temporal fluctuation and horizontal advection of $h$. Because these extra terms are on average nearly zero (Extended Data Fig. 3c), their inclusion does not affect $\overline{M}$. $\sigma_M$ instead increases by about 1.5 mm s$^{-1}$ owing to the pronounced variability in the temporal fluctuation term. As this term is not very robust, we use the more reliable equilibrium $M$ as our best estimate. The equilibrium $M$ is also robust at the 1-h scale of an individual circle (case '1h-scale').

Overall, Extended Data Fig. 2 makes us very confident in the robustness of our mass budget estimates because they only show a modest sensitivity to the various choices and because we can explain these sensitivities physically. Also, the independent ATR $M_{turb}$ estimates (Extended Data Fig. 2d) and the extra constraints on $E$ from our complementary analyses of the moisture and heat budgets in ref. [22] (dashed lines in Extended Data Fig. 2b) lend further credibility to our default estimation choices.

Next, we focus on the sensitivity of the key relationships between $M$, $C$ and $\mathcal{R}$ to a selected set of plausible estimation choices of $M$ and the different $C$ estimates from the ATR aircraft. Extended Data Fig. 5a shows that the positive correlation between $M$ and $C$ is notable for all parameter choices, and both the equilibrium $M$ and total $M'$. Furthermore, the negligible correlation between $M$ and $\mathcal{R}$ is also very robust.

Extended Data Fig. 5b further confirms that the default $M$ also has strong correlations with the three independent estimates of $C$ from the ATR aircraft. The same is true for the other estimation choices of $M$, with a small overall range of correlations of $0.52 < r_{M,C} < 0.73$. Correlations between $C$ and $\mathcal{R}$ are more variable between the different $C$ estimates and are in the range $0.12 < r_{\mathcal{R},C} < 0.63$. It is not surprising that the $C_{only}$ estimate that neglects contributions from drizzle has the strongest correlation with $\mathcal{R}$, as it mostly features passive clouds that are more affected by ambient humidity than the more active clouds that also include drizzle. Note that there is also a slight dependency of $r(\mathcal{R},C)$ on the $M$ estimates, as the cases 'h.parcel' and 'h.eps = 0.25' result in different $h$ and thus different heights at which $\mathcal{R}$ is evaluated.

The bottom panels of Extended Data Fig. 5 also confirm the robustness of the correlation coefficient of the multiple linear regression $\hat{C} = a_0 + a_M\widetilde{M} + a_\mathcal{R}\widetilde{\mathcal{R}}$ and the ratio of the standardized regression coefficients $a_M/a_\mathcal{R}$ to the $M$ estimation choices (Extended Data Fig. 5c) and the different $C$ estimates (Extended Data Fig. 5d). There is no configuration with $a_M/a_\mathcal{R} < 1$, indicating that $C$ is always more strongly coupled to $M$ than to $\mathcal{R}$ in the observations. Slightly larger values of $a_M/a_\mathcal{R}$ and smaller correlations are evident for the total $M'$.

Also, the standard deviation of $C$ ($\sigma_C$) is very similar for the different $C$ estimates that include drizzle (between 2.1% and 3.7%, with 3.1% being the $\sigma_C$ of the default BASTALIAS lidar-radar synergy product) and only slightly lower for the $C_{only}$ estimate (1.6%) when using the full sample. Variability is slightly reduced in the smaller sample that overlaps with the HALO flights, because it excludes two night flights with

larger cloudiness and two flights in dry environments with very small cloudiness ($\sigma_C$ of 1.7–2.4% for the $C$ estimates that include drizzle).

Overall, Extended Data Fig. 5 demonstrates the insensitivity of the observed relationships to a wide range of configurations. We therefore conclude that the relationships between mixing and cloudiness observed during EUREC[4]A are very robust.

## Data availability

All data used in this study are published in the EUREC[4]A database of AERIS (https://eurec4a.aeris-data.fr/, last accessed: 28 July 2022). We use v2.0.0 of the JOANNE dropsonde data[28] (https://doi.org/10.25326/246). The specific ATR datasets[31] used are the BASTALIAS product (https://doi.org/10.25326/316), the turbulence measurements[49] (https://doi.org/10.25326/128) and the PMA/Cloud composite dataset (https://doi.org/10.25326/237). The specific HALO datasets[29] used are cloud masks derived from WALES cloud-top height estimates (https://doi.org/10.25326/216), HAMP Cloud Radar (https://doi.org/10.25326/222) and specMACS (https://doi.org/10.25326/166), and the flight segmentation product (https://doi.org/10.5281/zenodo.4900003). From the BCO[15], we used ceilometer (https://doi.org/10.25326/367) and cloud radar data (https://doi.org/10.25326/55). From the RV Meteor[44], we used standard dship meteorological data for the EUREC[4]A Meteor cruise M161 (retrieved from http://dship.bsh.de/, last accessed: 28 June 2022), surface heat fluxes (https://doi.org/10.25326/312), ceilometer measurements (https://doi.org/10.25326/53) and cloud radar data (v1.1, https://doi.org/10.25326/164). We further used data from AutoNaut Caravela[52] (https://doi.org/10.25326/366) and 10-min air–sea flux data (v1.3, https://doi.org/10.25921/etxb-ht19) from the RV Ronald Brown[54]. Also, we used CLS Daily High Resolution Sea Surface Temperature maps (retrievable through the AERIS operational centre https://observations.ipsl.fr/aeris/eurec4a-data/SATELLITES/CLS/SST/, last accessed: 28 June 2022, or directly from https://datastore.cls.fr/catalogues/sea-surface-temperature-infra-red-high-resolution-daily), GOES-16 ABI SSTs from the ABI_G16-STAR-L3C-v2.7 product (https://doi.org/10.25921/rtf0-q898) and ERA5 (ref. [53]) reanalysis data. The CMIP5 and CMIP6 climate model outputs are available for download at https://esgf-node.llnl.gov. Source data are provided with this paper.

## Code availability

The scripts used for the analyses and other supporting information that may be useful for reproducing this study can be obtained from https://doi.org/10.5281/zenodo.7032765.

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

**Acknowledgements** We thank all the scientists, engineers, technicians, pilots and people from Barbados who made the EUREC[4]A data collection possible (see ref.[9] for the complete list). We acknowledge J. Delanoë and P. Chazette for making the horizontal radar-lidar cloud measurements on board the ATR possible, M. Lothon, P.-E. Brilouet and the SAFIRE team for making the turbulence measurements and ATR flights possible, M. Wirth and S. Gross for producing sub-cloud-layer height retrievals from the WALES lidar onboard HALO, C. Fairall and E. Thompson for insights about SSTs and surface fluxes, E. Siddle for sharing the AutoNaut Caravela data, I. Schirmacher for comparison with her RV Meteor surface flux estimates, J. Röttenbacher for making RV Meteor data available, I. Musat for providing climate model output and C. Hohenegger for feedback on the manuscript. The EUREC[4]A field campaign was financed by several international organizations[9]. The authors acknowledge an ERC Advanced Grant (EUREC[4]A grant no. 694768), base support from the Centre national de la recherche scientifique (CNRS) and the Max Planck Society (MPG), and additional support from the Max Planck Foundation and H2020 funding (CONSTRAIN grant no. 820829).

**Author contributions** R.V., S.B. and B.S. conceived the study, which was the primary objective of the EUREC[4]A campaign; R.V. performed the analyses and wrote the paper; J.V. processed the CFMIP output and helped define model diagnostics relevant to cloud feedbacks; A.L.A. helped with the entrainment rate estimate and the analysis of CFMIP model outputs; S.B. diagnosed the cloud fraction and mass flux estimates from ATR observations; G.G. contributed further JOANNE dropsonde and ERA5 products; and B.S. produced Fig. 1. All authors contributed to interpreting the results and writing the paper.

**Competing interests** The authors declare no competing interests.

**Additional information**
**Correspondence and requests for materials** should be addressed to Raphaela Vogel.

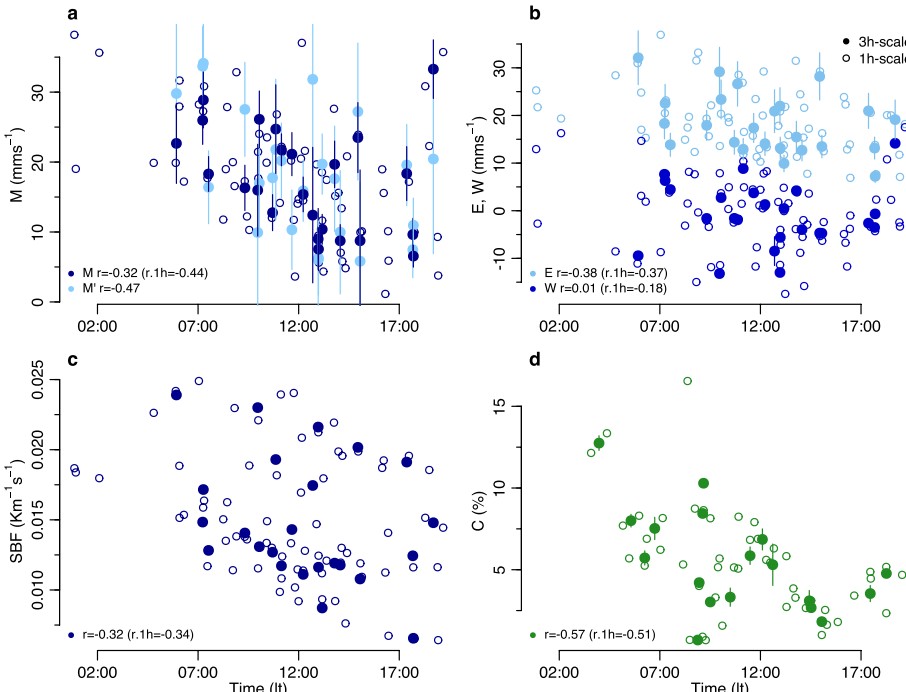

**Extended Data Fig. 1 | Diurnal cycles of key terms.** *M* and *M'* (**a**), *E* and *W* (**b**), surface buoyancy flux (**c**) and *C* (d) versus local time both at the 3-h scale (filled circles) and at the 1-h scale (open circles). The vertical bars show the estimation uncertainty at the 3-h scale (see Methods section 'Uncertainty estimation'). The correlation coefficients given in the legends represent the correlation between the individual terms and the time at the 3-h scale ('r') and at the 1-h scale ('r.1h' in brackets).

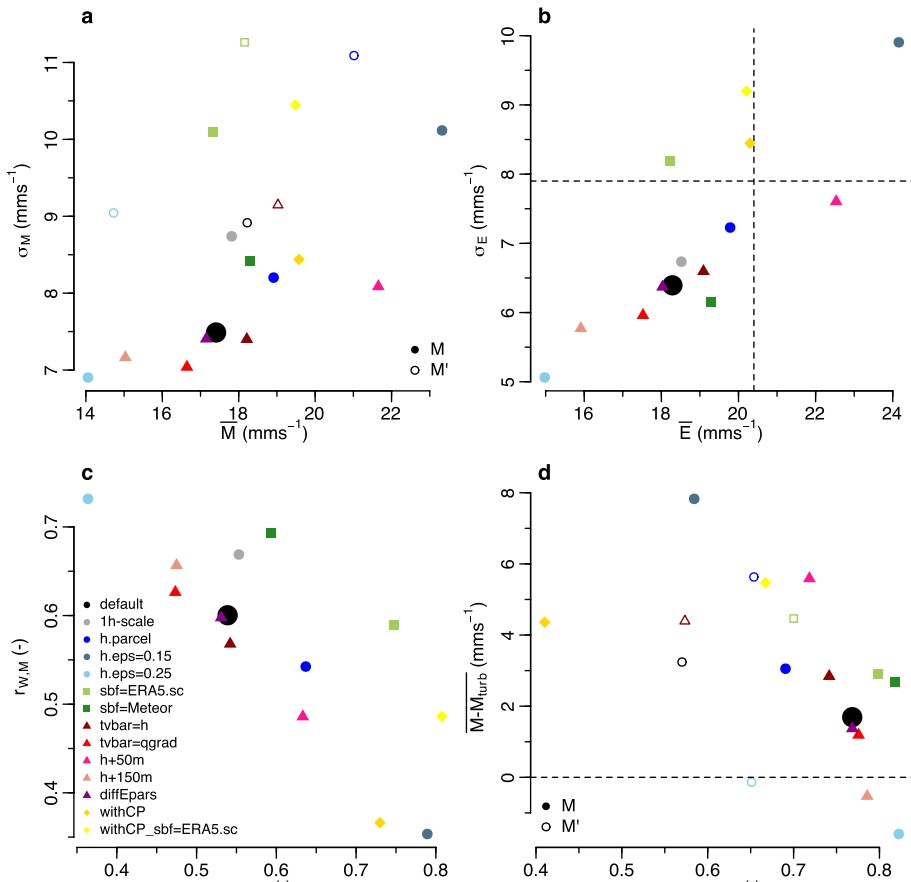

**Extended Data Fig. 2 | Influence of estimation procedure and parameter choices on mass budget estimates.** Campaign mean and standard deviation of $M$ (**a**) and $E$ (**b**). **c**, correlation coefficients between $M$ and $E$ ($r_{M,E}$) and $M$ and $W$ ($r_{M,W}$). **d**, Correlation and mean difference between $M$ and $M_{turb}$ from ATR turbulence measurements, for different configurations of the mass budget. Open symbols in **a** and **d** show the total $M'$. The dashed lines in **b** show the mean and standard deviation of $E$ from ref. [22] and the zero line in **d**. See the Methods section 'Robustness of observational estimates' for details.

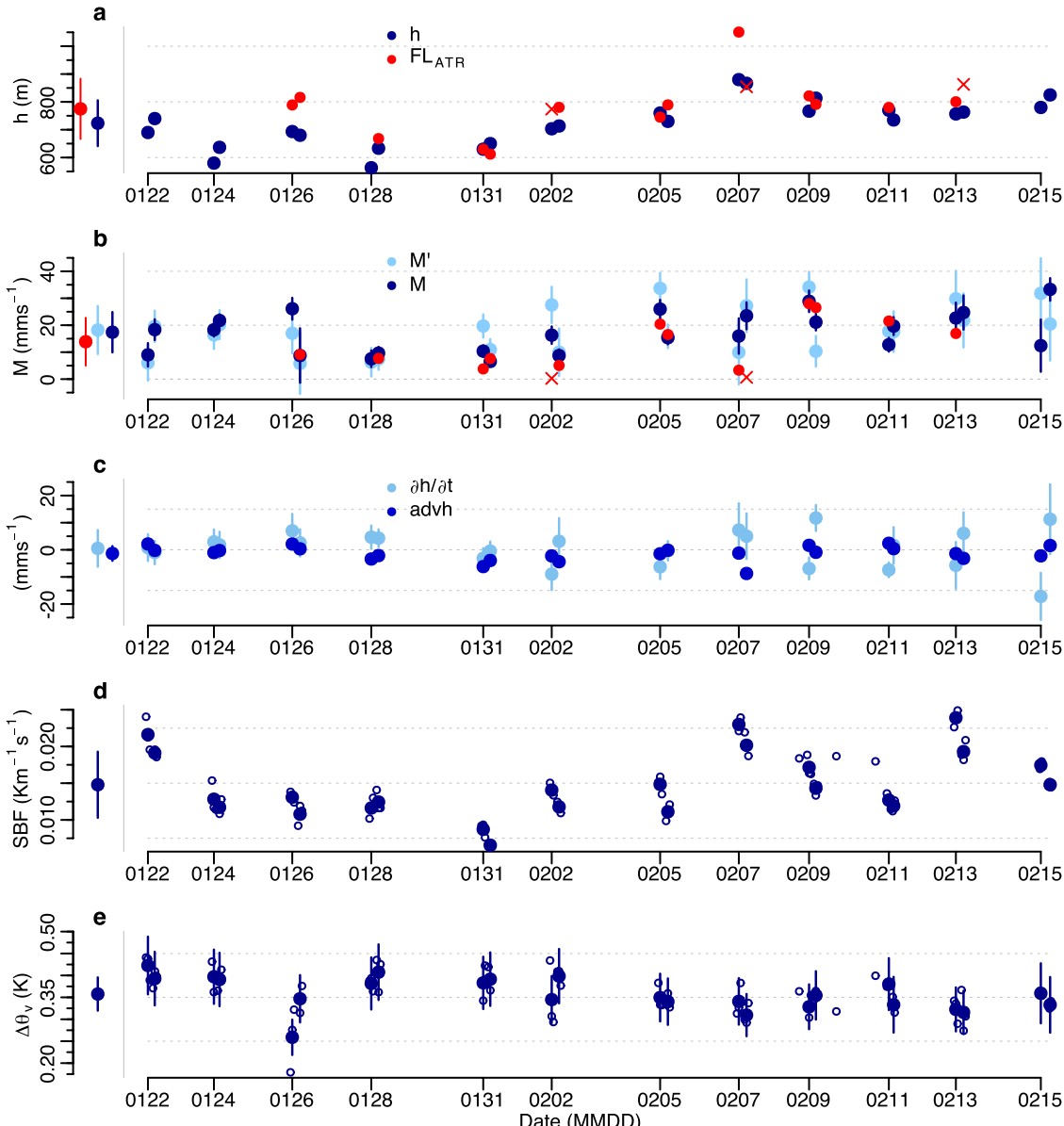

**Extended Data Fig. 3 | Time series of other mass budget terms.** Shown are $h$ and the flight level of the ATR aircraft (**a**), the equilibrium $M$, the total $M'$ and the $M_{turb}$ from ATR turbulence measurements (**b**), the temporal fluctuation and advection terms (**c**), the surface buoyancy flux (**d**) and the $\Delta\theta_v$ (**e**). The vertical bars show the estimation uncertainty at the 3-h scale (see Methods section 'Uncertainty estimation') and the small open circles show the 1-h scale. The 'X' markers in **a** and **b** indicate the data that are excluded in the correlations owing to inconsistent sampling between the two aircraft. The campaign mean ± 1σ is shown on the left side of each panel.

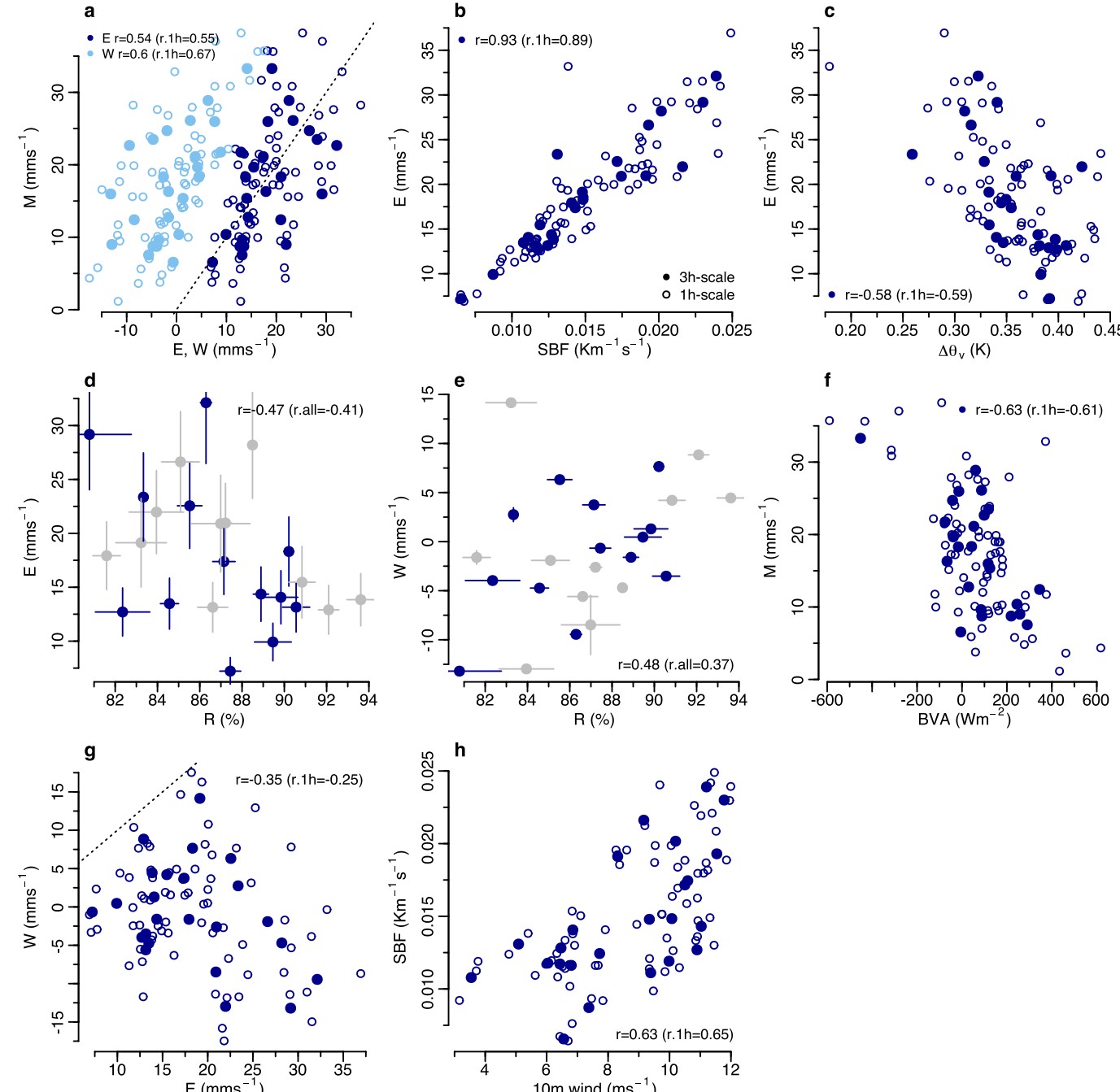

**Extended Data Fig. 4 | Relationships of other key terms. a**, $E$ and $W$ versus $M$. **b**, Surface buoyancy flux versus $E$. **c**, $\Delta\theta_v$ versus $E$. **d**, $\mathcal{R}$ versus $E$. **e**, $\mathcal{R}$ versus $W$. **f**, BVA mixing indicator versus $M$. **g**, $E$ versus $W$. **h**, 10-m wind speed versus surface buoyancy flux. **a**–**c**, **f**–**h** show both the 3-h scale (filled circles) and the 1-h scale (open circles, with the corresponding correlation coefficient denoted as 'r.1h'). A dotted 1:1 line is shown in **a** and **g**. In **d** and **e**, the error bars represent the estimation uncertainty for $E$ and $W$ and the sampling uncertainty for $\mathcal{R}$ (see Methods). The correlations in **d** and **e** are given both for the sample with consistent sampling among the HALO and ATR aircraft (blue points, as used for the correlations in Figs. 2 and 3) and for the entire sample of the HALO aircraft (including the grey points that represent the three data points marked with 'X' in Fig. 3 and eight other data points when ATR was not flying. The corresponding correlation coefficient is denoted as 'r.all').

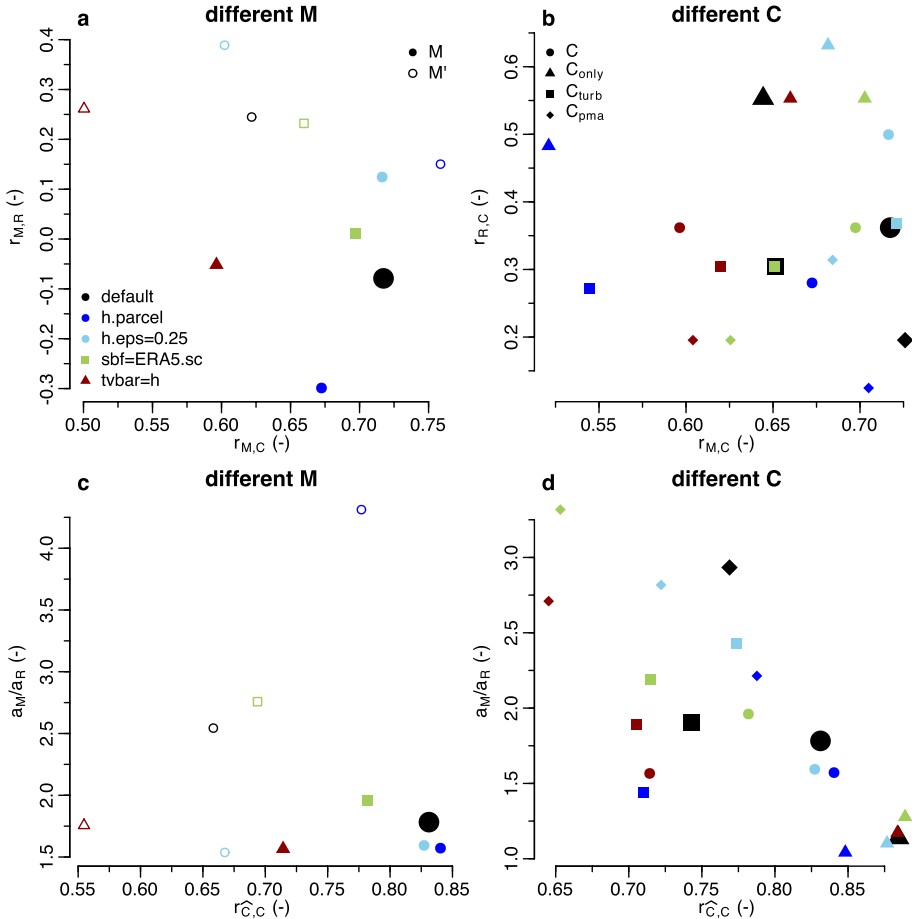

**Extended Data Fig. 5 | Influence of different *M* and *C* estimates on key relationships.** Correlation coefficients *r* of *M* and *C* ($r_{M,C}$) and *M* and $\mathcal{R}$ ($r_{M,\mathcal{R}}$) (**a**) and *M* and *C* ($r_{M,C}$) and $\mathcal{R}$ and *C* ($r_{\mathcal{R},C}$) (**b**). **c**,**d**, Correlations of the reconstructed $\hat{C} = a_0 + a_M \widetilde{M} + a_\mathcal{R} \widetilde{\mathcal{R}}$ and the observed *C* ($r_{\hat{C},C}$), as well as the ratio of the standardized regression coefficients $a_M/a_\mathcal{R}$. **a** and **c** also show the relationships for the total *M'* (open symbols), whereas **b** and **d** show the relationships for different estimates of *C* (different symbols). See details in Methods section 'Robustness of observational estimates'.

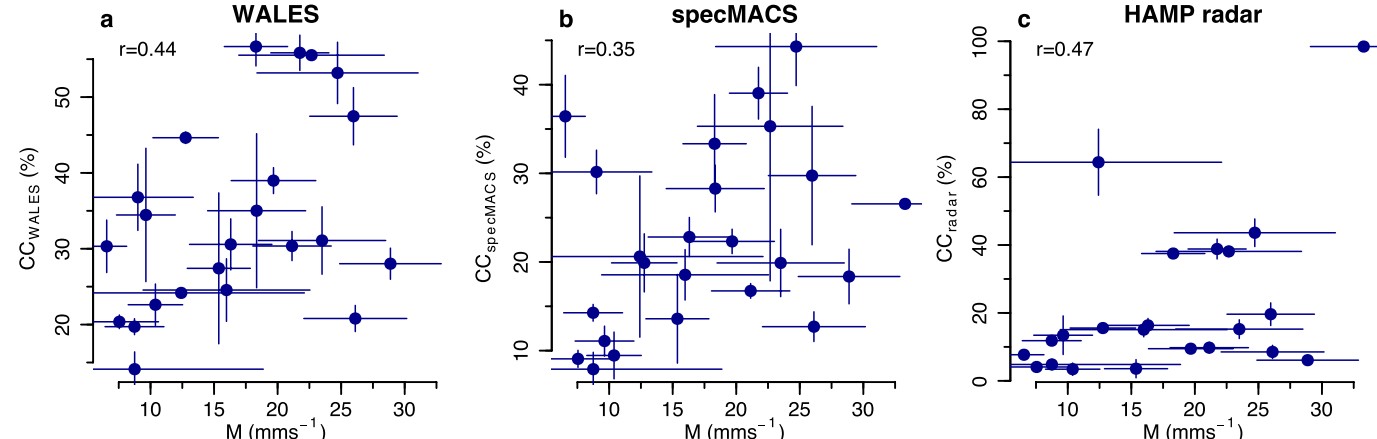

**Extended Data Fig. 6 | Relationship of *M* with three estimates of the total projected cloud cover.** Cloud cover (CC) from WALES backscatter lidar (**a**), hyperspectral imager specMACS (**b**) and HAMP cloud radar (**c**) on board HALO. The error bars represent the sampling uncertainty (for the CC estimates) and the estimation uncertainty (for *M*; see Methods section 'Uncertainty estimation').

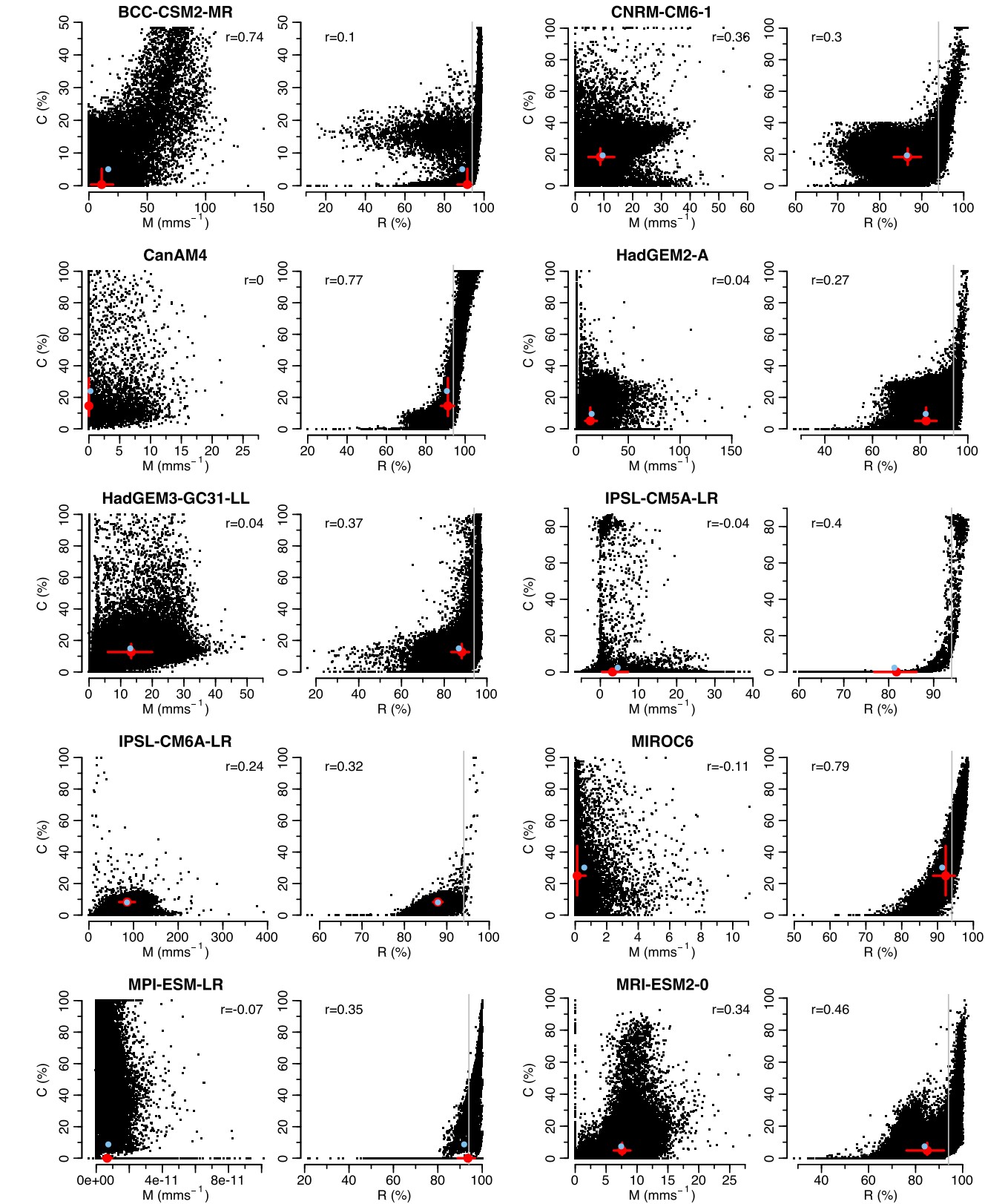

**Extended Data Fig. 7 | Individual relationships of *C*, *M* and $\mathcal{R}$ for climate models.** Relationships among individual 3-h *C* and *M* (first and third columns) and *C* and $\mathcal{R}$ (second and fourth columns) for all ten climate models. The red and blue points represent the median and mean of the respective variables, and the red lines extend from the 25th to the 75th quartile. The grey vertical line in the $\mathcal{R}$ panels shows the 94% $\mathcal{R}$-threshold.

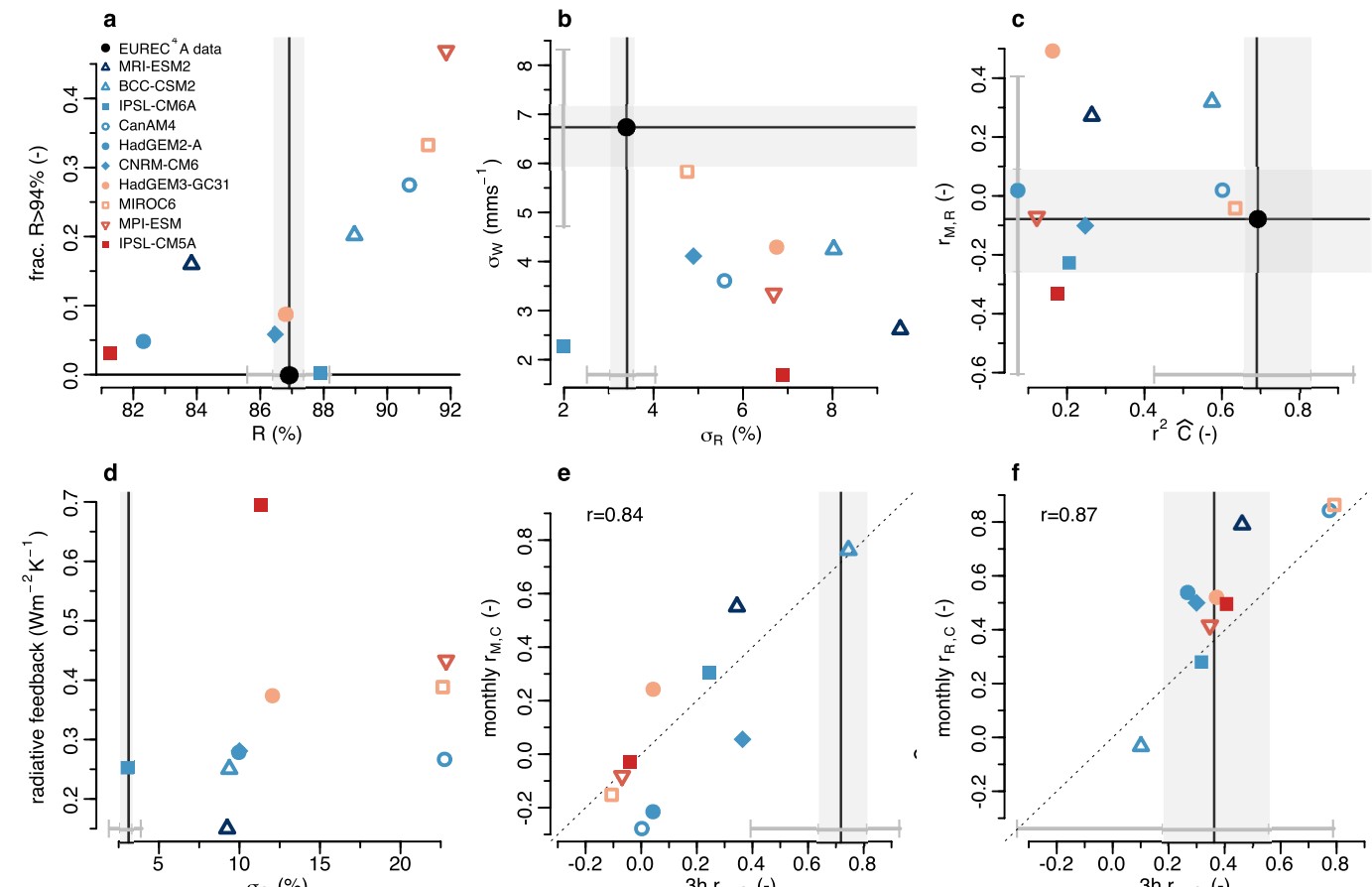

**Extended Data Fig. 8 | Comparison of other variables and relationships in climate models against the EUREC⁴A data. a**, Mean $\mathcal{R}$ and fraction of stratocumulus-like conditions with $\mathcal{R} > 94\%$. **b**, Standard deviation of $\mathcal{R}$ and $W$ ($\sigma_{\mathcal{R}}$ and $\sigma_W$). **c**, $r^2$ of multiple linear regression $\hat{C} = a_0 + a_M \widetilde{M} + a_{\mathcal{R}} \widetilde{\mathcal{R}}$ and correlation coefficient of $M$ and $\mathcal{R}$. **d**, Standard deviation of $C$ ($\sigma_C$) and thermodynamic component of the cloud feedback $\Delta CRE / \Delta T_s$, as well as the 3-h and monthly correlations of $M$ and $C$ (**e**) and $\mathcal{R}$ and $C$ (**f**). **e** and **f** also show the inter-model correlation coefficients of the respective variables and the 1:1 line (dotted line). As in Fig. 4, the models are coloured in bins of feedback strength and open symbols indicate models with frequent stratocumulus (defined as having $\mathcal{R} > 94\%$ more than 15% of the time). The observational uncertainty range is shown in grey, with the shading representing the 25th to 75th quartile and the grey bars indicating the 95% confidence interval of bootstrapped values. HadGEM2-A is not shown in **b** owing to the absence of $W$ output.