## [Peer Review File · Nature]

Manuscript Title: Strong cloud-circulation coupling explains weak trade cumulus feedback

Redactions – unpublished data

Parts of this Peer Review File have been redacted as indicated to maintain the confidentiality of unpublished data

Reviewer Comments & Author Rebuttals

Reviewer Reports on the Initial Version:

Referees' comments:

Referee #1 (Remarks to the Author):

Summary:

The authors use measurements from the EUREC4A field campaign to constrain the role of lower tropospheric mixing on cloudiness. Specifically the goal is to test the strength of mixing-desiccation on cloud fraction in the real world, which is observed to be a controlling mechanism in many GCMs. They find in the EUREC4A measurements that entrainment (E) and meso-scale motions (vertical velocity, W) contribute equally to lower tropospheric mixing (mass flux, M). Because entrainment and meso-scale vertical velocity have opposing effects on humidity, increased mixing does not lead to desiccation. The authors are able to use these aircraft measurements then to refute the mixing-desiccation hypothesis. The authors then use their derived process-based constraint as an emergent constraint on GCMs. They find that in GCMs parameterized clouds are more thermodynamically than dynamically controlled (opposing the observations). They show that all models (with relevant CFsites output) with large positive radiative feedback ($> 0.3 \text{ W/m}^2/\text{K}$) have an active mixing-desiccation mechanism, therefore constraining the feedback.

Overall, I thought this paper was quite well written and easy to read. The analysis is very careful with many detailed checks. The supplement shows that the results are robust to many alternative ways of estimating important quantities from the in situ measurements. I recommend accepting this paper after minor revisions.

Major comments:

1. L51: It would be nice to include a mathematical description of how the relative humidity enters the mixing-desiccation mechanism discussion. It was confusing just reading the text at first to understand why R was brought up in this line when it wasn't discussed again until the next page. It is discussed later on that depending on the trade-off between E and W, increasing M can either lead to decreasing R or little change in R.

Maybe formalizing this (combining Eq. 1 with the $C \sim M$ equation) would be useful?

Also, I found Extended Data Fig. 1 very instructive. It would be great if there is space for it in the main text.

2. Throughout, the terminology of "circle" vs. "circling" was quite confusing to remember, especially when the "1h" and "3h" prefixes were dropped.

Minor comments:

- L32: the phrase “large statistical weight” here is odd.
- Fig. 1 is not as useful as it could be. It’s hard to read the y-values of points.
- Fig. 2a: Is the dotted line the 1:1 line?
- Fig. 2: The caption about the grey excluded points is hard to understand without reading L339-343 in the methods. More detail can be added here to explain “inconsistent sampling of mesoscale variability”
- L140: “data is therefore”  “data are therefore”
- L147: “(M)”  “(larger M)”
- L148: “(decreasing C)”  “(smaller C)”
- L165: How does using the regimes defined by observations (Myers et al. 2021) impact the results? If ShCu regimes were diagnosed from the models themselves, would the relationships investigated between M and C or the calculated radiative effect be more aligned with the observations? I’m curious how much is a problem of the models getting the location of the different cloud regimes wrong vs. the models having unrealistic physics in the ShCu regime.
- Fig. 3: My take away from Fig. 3 was the sort of emergent constraint discussed in L204-L214. It would be interesting if the figure could be redone to show this explicitly, like plotting $r_{\{M,C\}}$ against the radiative feedback directly?
- L327: Put BCO in parentheses.
- L349: “from12” needs space
- L350: 10% larger RH seems quite significant. How does this compare with ERA5?
- L414: Why is the SHF overestimated in ERA5 by 64%? This seems very sizeable!
- L429: I think it’s more confusing to reproduce Eq 1 in line here than just reference it.
- L738: “Panel b” and other such references to figures without using the figure number should be avoided. Change to “Extended Data Fig. 6b” for example.

Referee #2 (Remarks to the Author):

This paper tests the hypothesis used to explain the high climate sensitivity of trade wind cumulus clouds in climate models (i.e., a decrease in cloud-base cloudiness due to increased lower-tropospheric mixing causes the Trade cumulus feedback) by observationally examining the magnitudes and correlations of key variables. The results show that this hypothesis is not supported because cloudiness is controlled not only by mixing but also, more significantly, by mesoscale vertical motion. I would say that the close relationship between cloudiness and mesoscale vertical motion is not a new finding (e.g., Bretherton and Blossey 2017, <https://doi.org/10.1002/2017MS000981> ; George et al. 2021, <https://doi.org/10.1175/JAS-D-20-0335.1>). However, this manuscript is an excellent demonstration quantifying the linkages between key variables that explain different climate sensitivities across climate models. There are no obvious flaws that would prohibit publication. The conclusions are original and will be of interest to many. Below are a few specific comments.

1) Do temporal and spatial scales affect the conclusions? I am curious as to what spatial and

temporal scales the study focuses on. There would be a lot of "mesoscale" motion between cloud and flight circle scales. Also, most mesoscale motions are not available in climate models due to coarse grid spacing. Do mesoscale motions become more important as the target scale decreases? Could thermodynamic effects become more important as the target scale increases and mesoscale motions are averaged? Scale effects need to be considered for fair comparison and discussion.

2) There are some minor issues in the uncertainty discussion. There is an overemphasis in this paper on the first measurements of key variables and relationships. It feels to me that those are estimates based on observations, not direct measurements. For example, the mass flux M is based on the residuals of the budget equation and the entrainment E is based on a fixed portion of the surface flux. Cloudiness C from aircraft remote sensors is based on a novel approach but has inherent uncertainties (e.g., Cloudiness near cloud base of shallow cumulus clouds can be highly dependent on the height, and small changes in flight level can make a large difference; Droplets are often too small to be observed by radar).

Referee #3 (Remarks to the Author):

I have reviewed "Strong cloud-circulation coupling explains weak trade cumulus feedback" by Vogel et al. The authors present an analysis of covariability of convective mass flux, large-scale vertical velocity, entrainment, relative humidity, and cloud fraction in the tradewind cumulus cloud regime. The two main conclusions are that observational results are inconsistent with the mixing-desiccation hypothesis, reducing the plausibility of a strongly positive cloud feedback in tradewind cumulus. A somewhat related conclusion is that GCMs overestimate the sensitivity of cloud fraction to thermodynamic (RH) changes and underestimate its sensitivity to circulation.

These are highly relevant and timely conclusions of interest to a broad range of scientists, not least because they contribute to the intense discussion of potentially high climate sensitivity in the wake of CMIP6. As part of this discussion, a great deal of process realism analysis has been done on midlatitude clouds (e.g., Tan et al. 2016, McCoy et al. 2020, Mulmenstadt et al. 2021, Fraser et al. 2022), but the equally or perhaps even more important tropical shallow cumulus regime (Zelinka et al. 2020, Myers et al. 2021) has received much less process attention. It is refreshing to see this well designed study address this important cloud type.

As far as I can tell, the analysis is sound. My main concern is a nagging doubt about using present-day (monthly) variability to make a direct statement about a very low-frequency perturbation like anthropogenic warming. I would feel better if we had a modeling system that allowed us to make the link across time scales. As the authors demonstrate, the models we do have are all quite far removed from process realism, so I am not sure how far dividing them into "very unrealistic" and "even more unrealistic" can get us in terms of estimating what is realistic. This is certainly not a criticism of this study, more a criticism of the tools we currently have available to us. Or perhaps it is a lengthy way of saying that the part of the paper I found of greatest relevance was the refutation of the mixing-desiccation storyline, not the reading of the CMIP tea leaves (Fig. 3c).

The comments below are offered as constructive criticism. If they are easy to implement, I think they may assist the reader; if they are time-consuming, then I would rather that the authors ignore these suggestions than that they hold up publication.

* My main substantive question concerns the entrainment calculation that enters into the calculation of M . The entrainment estimate is a parameterization based on more readily observable quantities and, as such, relies on parameter estimation. Did the authors account for the possibility that the parameter estimate itself could depend on the mesoscale environment? I ask because such a parameter dependence could spuriously produce or suppress covariabilities between M and W , C , or R . Since the authors' conclusions rely on covariability between C and $\{W, R, M\}$, spurious covariability introduced by the entrainment parameterization would be problematic.

* I was slightly perplexed that the analysis is presented as the *raison d'être* of the EUREC4A campaign and synthesizes many of the data streams collected during the campaign, but that the author list and acknowledgments do not reflect the many scientists and engineers who contributed to data collection. I suppose citing the campaign description papers is an alternate form of acknowledgment, and maybe this is a cultural difference.

* The authors describe the analysis as a "process-based constraint". However, I could not find an explicit statement of which process(es) the authors claim to be constraining. Entrainment? Organization? Convection? The "convection" process in GCMs? I am not disagreeing about the label, but I think more specificity would avoid the buzzwordy taste that "process-oriented" is starting to acquire.

* On the presentation, I found the use of "circle" and "circling" to mean different things quite confusing.

* Further on the presentation, I don't love that M is frequently referred to as "mass flux" when its units are velocity.

* l. 36: "make up for"  "make up"

Author Rebuttals to Initial Comments:

Response to referees

Nature manuscript 2022-05-07332 entitled "Strong cloud-circulation coupling explains weak trade cumulus feedback" by Vogel et al. 2022.

We thank the editor and the three referees for their constructive feedback. The referees raise a few important issues that we are happy to address in our revision. The specific comments of the referees and the editorial changes requested are addressed below, with our responses inserted in green font.

Referees' comments:

Referee #1 (Remarks to the Author):

Summary:

The authors use measurements from the EUREC⁴A field campaign to constrain the role of lower tropospheric mixing on cloudiness. Specifically the goal is to test the strength of mixing-desiccation on cloud fraction in the real world, which is observed to be a controlling mechanism in many GCMs. They find in the EUREC⁴A measurements that entrainment (E) and meso-scale motions (vertical velocity, W) contribute equally to lower tropospheric mixing (mass flux, M). Because entrainment and meso-scale vertical velocity have opposing effects on humidity, increased mixing does not lead to desiccation. The authors are able to use these aircraft measurements then to refute the mixing-desiccation hypothesis. The authors then use their derived process-based constraint as an emergent constraint on GCMs. They find that in GCMs parameterized clouds are more thermodynamically than dynamically controlled (opposing the observations). They show that all models (with relevant CFsites output) with large positive radiative feedback ($> 0.3 \text{ W/m}^2/\text{K}$) have an active mixing-desiccation mechanism, therefore constraining the feedback.

Overall, I thought this paper was quite well written and easy to read. The analysis is very careful with many detailed checks. The supplement shows that the results are robust to many alternative ways of estimating important quantities from the in situ measurements. I recommend accepting this paper after minor revisions.

Major comments:

1. L51: It would be nice to include a mathematical description of how the relative humidity enters the mixing-desiccation mechanism discussion. It was confusing just reading the text at first to understand why R was brought up in this line when it wasn't discussed again until the next page. It is discussed later on that depending on the trade-off between E and W, increasing M can either lead to decreasing R or little change in R.

Maybe formalizing this (combining Eq. 1 with the $C \sim M$ equation) would be useful?

Also, I found Extended Data Fig. 1 very instructive. It would be great if there is space for it in the main text.

We agree that the role of the relative humidity did not receive enough attention in the introduction and formalizing its role in the mechanism is useful. In the revised version, we directly included R in the $C \sim M$ relationship, because that's essentially what the mixing-desiccation mechanism contends: $C \propto R \propto M^\beta$, mixing induces drying, which induces the cloud reduction. We also modified the sentence in L60-62 to include R explicitly:

*"Yet active clouds represent only half of the total C [18,19] and the lifetime and variability of passive clouds, such as the detritus of decaying clouds, might be more sensitive to **R** and mixing-induced drying of their environment than to M."*

A key result of our study is then that we do not see much of this $R \propto M$ in the observations, as the two are hardly correlated. So this emphasizes a more direct link between $C \sim M$, and R is just a secondary predictor. We thus modified the sentence in L139f saying: *"The EUREC⁴A data are therefore in line with a more direct relation $C \propto M^\beta$ and a $\beta > 0$ (Extended Data Fig.~1b)."*

We also agree that moving the schematic in Extended Data Fig.1 to the main text would be helpful for many readers. We will discuss this with the editor to see if there is space.

2. Throughout, the terminology of "circle" vs. "circling" was quite confusing to remember, especially when the "1h" and "3h" prefixes were dropped.

We used this circle vs. circling nomenclature as it was introduced in the EUREC⁴A overview

paper (Stevens et al. 2021) and has become standard in the following (e.g. Konow et al. 2021, Albright et al. 2022). Nevertheless, we agree that it can be confusing and thus completely drop the 'circle / circling' nomenclature and only keep the timescale prefixes. So the (3h-)circling scale becomes the 3h-scale, and the (1h-)circle scale becomes the 1h-scale. When referring to a specific 3h "circling" period, we refer to it as a '3h-circle set(s)'.

As the airmasses are advected by ~30km per hour (at the campaign-mean windspeed of ~9 m/s at 1km height), the spatial sampling of the ~220km diameter circle doesn't differ substantially between the 1h and 3h timescales, such that it makes sense to focus the nomenclature on the time rather than the space scale.

Minor comments:

- L32: the phrase "large statistical weight" here is odd. We changed the sentence to: "*Due to their large geographical extent, small errors in predicting the way trade cumuli respond to warming can have a large effect on the global radiative budget.*"
- Fig. 1 is not as useful as it could be. It's hard to read the y-values of points. We added thin horizontal reference lines to all panels to ease reading of the y-values in Fig. 1. We also added such reference lines to Extended Data Fig. 4.
- Fig. 2a: Is the dotted line the 1:1 line? Yes. We added '*The dotted line in a is the 1:1 line.*' in the figure caption.
- Fig. 2: The caption about the grey excluded points is hard to understand without reading L339-343 in the methods. More detail can be added here to explain "inconsistent sampling of mesoscale variability"
We also discuss this sampling issue in the context of Fig. 1d in L111-L114. To clarify this point, we therefore added a reference to both the Methods section and Fig. 1d, and simplified the sentence in the Figure legend by removing the term 'mesoscale variability'. The new sentence is: "*The grey x markers represent data that are excluded in the correlations due to inconsistent sampling between the two aircraft (see Fig. 1d and Methods).*"
- L140: "data is therefore"  "data are therefore" Changed
- L147: "(M)"  "(larger M)" Changed
- L148: "(decreasing C)"  "(smaller C)" Changed
- L165: How does using the regimes defined by observations (Myers et al. 2021) impact the results? If ShCu regimes were diagnosed from the models themselves, would the relationships investigated between M and C or the calculated radiative effect be more aligned with the observations? I'm curious how much is a problem of the models getting the location of the different cloud regimes wrong vs. the models having unrealistic physics in the ShCu regime. It is indeed not obvious if the chosen cfsites locations are representative of shallow trade cumulus regimes, or if the models are biased in terms of their large-scale circulation and the cfsites end up being in more stratocumulus-like regimes. We addressed this point in a submitted paper by Vial, Albright, Vogel, Musat and Bony ('The daily cycle constrains the realism of simulated trade cumulus feedbacks to warming'), where we checked the seasonally averaged range of values of ω_{700} and EIS as used in the Myers et al. 2021 regime partitioning (see their Fig. S10 shown below).

We find that the EIS is well within the trade-cumulus regime range defined by Myers et al. 2021 (EIS < -1K), while the average ω_{700} is slightly larger than the Myers range of $0 < \omega_{700} < 15$ hPa/d.

This difference in ω_{700} is due to Myers et al. 2021 defining the regimes on an annual basis, which yields lower ω_{700} than if winter months only are considered. The pronounced shift in the circulation regime from moderately subsiding motion in winter to moderately rising motion in summer near Barbados mirrors the seasonal migration of the intertropical convergence zone (see Fig. 4 of Brueck et al. 2015 for ERA-interim and Fig. S1 of Medeiros and Nuijens 2016 for CMIP models). Also, the average ω_{700} of the EUREC⁴A dropsondes is well within the model range (mean 23.3 hPa/d and median 28.2 hPa/d). We therefore conclude that the models are getting the cloud regime right, but have unrealistic physics in the trade cumulus regime.

Redacted

Vial et al. (in review) also confirms that the cloud fraction and mass flux changes with warming are quasi the same whether we use the hourly cfsites output or an alternative ω_{500} regime classification with monthly Amon output, and the same conclusion likely also holds for the Myers et al. 2021 regime partitioning. The trade cumulus radiative feedback is computed from the monthly output for a broader region using all points falling into ω_{700} and EIS regime of Myers et al. 2021, and thus a priori consistent with this definition.

We added a sentence to clarify this point in the revised manuscript (L616): “*We verified that the cfsites used fall into the trade cumulus regime as defined by Myers et al. (2021).*”

• Fig. 3: My take away from Fig. 3 was the sort of emergent constraint discussed in L204-L214. It would be interesting if the figure could be redone to show this explicitly, like plotting $r_{\{M,C\}}$ against the radiative feedback directly?

We did plot the correlation of M and C versus the radiative feedback and do find a pronounced negative relationship between feedback strength and M~C correlation (see Fig. 2 below). Fitting a linear regression to this data, the M~C correlation explains 30% of variability in the radiative feedback (not shown). When excluding two of the most ‘problematic’ models, IPSL5 and BCC,

the fit would improve further, explain 60% of the variability in the feedback, and yield a ‘most-likely’ observationally-constrained feedback of about 0.05-0.1 W/m²/K, similar to the value derived from interannual variability in Myers et al. 2021.

There are, however, several reasons why we don’t include such a figure in the paper. Echoing the general comment of referee 3, all models are quite far from reality, such that making a prediction based on a small set of models that all have their issues might not be useful. Emergent constraints are also often associated with “data mining” issues, showing much smaller explanatory power for out-of-sample data such as a new model ensemble (Caldwell et al. 2018, Schlund et al. 2020). Furthermore, focusing just on the M~C correlation does not account for the complexity of the processes involved, as e.g. BCC turns out to be way-off from the observations when also accounting for the RH~C relationship.

We therefore don’t try to provide an explicit constraint on the feedback strength. What we can say that the models predicting a strong positive feedback are unrealistic, but a number of models predicting a weak feedback are also unrealistic. A model will be assessed to be realistic in its trade cumulus feedback when it will at least reproduce the C~M and C~RH relationships in agreement with observations. We emphasize again at the end of the manuscript that testing (and refuting) the idea of the mixing-desiccation mechanism is the key point of the manuscript, and assessing what the climate models do is a secondary objective.

Fig. 2: Correlation coefficients between M and C ($r_{M,C}$) versus the thermodynamic component of the trade cumulus radiative feedback. The models are colored in bins of feedback strength. Open symbols refer to models with frequent stratocumulus. The grey shading represents the 25th to 75th quartile and the grey bars the 95%-CI of bootstrapped observational values.

• L327: Put BCO in parentheses. **Changed**

• L349: “from12” needs space **Changed**

• L350: 10% larger RH seems quite significant. How does this compare with ERA5?

The EUREC⁴A dropsonde-circle mean RH at 850hPa is also about 8% larger than the 10-year ERA5 average (see Fig. 3 below). The same difference is found for the ERA5 Jan-Feb 2020 average, when EUREC⁴A took place. We don’t want to go into detail regarding this difference, and add in L375 of the revised manuscript (here in bold):

“The mean meteorological conditions during the EUREC⁴A campaign, as sampled by the dropsondes, also correspond well to the average January-February conditions from 12 years of data from the ERA-Interim reanalysis (Brueck2015, their Fig.~5), albeit with a 10% larger 850hPa relative humidity during EUREC⁴A (an ~8% relative humidity offset is also found compared to the 2013-2022 average in ERA5, not shown).”

Fig. 3: Yearly averaged 850hPa RH for Jan-Feb 2013-2022 from ERA5, averaged over a square area enclosing the EUREC4A circle. Error bars represent ± 1 sd of hourly averages. EUREC4A campaign-mean 850 hPa RH from all the 1h circles, with error bars representing ± 1 sd of 1h circle-means. The thin dotted lines represent the 10-year (ERA5) resp. EUREC4A campaign mean.

- L414: Why is the SHF overestimated in ERA5 by 64%? This seems very sizeable!

Yes, the relative difference is sizeable. Because the SHF is generally small in absolute terms (6 W/m² for the dropsonde-derived flux compared to 9.8 W/m² for the ERA5 flux), it is still ‘just’ a ~3.8 W/m² overestimation. The latent heat flux is overestimated in ERA5 by 14.7 W/m², which is 9% larger compared to the dropsondes. An 11-year long comparison of ERA-interim (and 2 other reanalyses) with NTAS buoy measurements in Bigorre and Plueddemann (2021) shows that the predecessor of ERA5 strongly overestimates the SHF in the Barbados region (Jan-Feb averages of ~12.5 W/m² for ERA-interim compared to ~6.5 W/m² for NTAS ASIMET buoy measurements).

We added in the revised version (here in bold): “*The ERA5 fluxes, however, overestimate the surface buoyancy flux compared to the sonde-derived flux by 25%, which is mostly due to the overestimation of the sensible heat flux by 64% relative to the observations (9.8 Wm⁻² and 6.0 Wm⁻² for ERA5 and dropsondes, respectively). A strong overestimation of the sensible heat flux compared to buoy measurements in the region is also present in the predecessor ERA-interim reanalysis (Bigorre and Plueddemann, 2021).*”

- L429: I think it’s more confusing to reproduce Eq 1 in line here than just reference it. With the benefit of hindsight, we agree and omit reproducing eq 1 in the revised manuscript.
- L738: “Panel b” and other such references to figures without using the figure number should be avoided. Change to “Extended Data Fig. 6b” for example. Changed throughout the paper.

Referee #2 (Remarks to the Author):

This paper tests the hypothesis used to explain the high climate sensitivity of trade wind cumulus clouds in climate models (i.e., a decrease in cloud-base cloudiness due to increased lower-tropospheric mixing causes the Trade cumulus feedback) by observationally examining the magnitudes and correlations of key variables. The results show that this hypothesis is not supported because cloudiness is controlled not only by mixing but also, more significantly, by mesoscale vertical motion. I would say that the close relationship between cloudiness and mesoscale vertical motion is not a new finding (e.g., Bretherton and Blossey 2017, <https://doi.org/10.1002/2017MS000981> ; George et al. 2021, <https://doi.org/10.1175/JAS-D-20-0335.1>). However, this manuscript is an excellent demonstration quantifying the linkages between key variables that explain different climate sensitivities across climate models. There are no obvious flaws that would prohibit publication. The conclusions are original and will be of interest to many. Below are a few specific comments.

1) Do temporal and spatial scales affect the conclusions? I am curious as to what spatial and temporal scales the study focuses on. There would be a lot of "mesoscale" motion between cloud and flight circle scales. Also, most mesoscale motions are not available in climate models due to coarse grid spacing. Do mesoscale motions become more important as the target scale decreases? Could thermodynamic effects become more important as the target scale increases and mesoscale motions are averaged? Scale effects need to be considered for fair comparison and discussion.

- Scale definition:

This is an important point that we are happy to address in more detail in the revised manuscript. We use measurements made in the 220km-diameter 'EUREC⁴A' circle. We focus on the 3-hourly timescale of three consecutive circles, but also show results for the 1-h timescale of a single circle. In response to the referee comments, we clarified our scale definition. The (3h-) circling scale becomes the 3h-scale, and the (1h-) circle scale becomes the 1h-scale. As the airmasses are advected by ~30km per hour (at the campaign-mean windspeed of ~9 m/s at 1km height), the spatial sampling of the ~220km diameter circle doesn't differ substantially between the 1h and 3h timescales, such that it makes sense to focus the nomenclature on the time rather than space scale.

With EUREC⁴A we targeted the lower end of the meso-alpha scale (200-300km, Orlanski 1975) because this is the relevant scale of cloud processes for a trade cumulus ensemble and also the scale that convective parameterizations target. The 200-300km scale is in between the $O(1\text{ km})$ scale of individual clouds and the synoptic scale $O(1000\text{ km})$, and is also associated with the emergence of the four prominent cloud organization patterns of Stevens et al. 2019.

- Scale sensitivity:

When we developed the method to estimate M as a residual of the mass budget with ICON large-eddy simulations (LES) in Vogel et al. 2020, we found a limited sensitivity of the relative roles of W and E at explaining variability in M when the domain size was quadrupled (from $1^\circ \times 2^\circ$ to $2^\circ \times 4^\circ$). We thus noted in Vogel et al. 2020 that "The role of E in explaining the diurnal cycle of M is insensitive to halving the resolution and quadrupling the domain size. Also the variability in W continues to explain most of the (somewhat lower) day-to-day variability of M on a 4-fold larger domain." In the LES study, however, W contributed disproportionately more to variability in M compared to E , because we were using a simpler E formulation due to coarse vertical grid spacing of the LES. For the EUREC⁴A data, the correlation of W and M is only slightly larger than the correlation of E and M .

From the EUREC⁴A data itself, we can compare variability at the 3h-scale (3 circles) with variability at the 1h-scale (1 circle). ED Fig. 5a shows that the correlation between W and M is slightly larger at the 1h-scale compared to the 3h-scale ($r_{W,M\ 3h} = 0.60$ and $r_{W,M\ 1h} = 0.67$), while

they are essentially the same for E and M ($r_{E,M,3h} = 0.54$ and $r_{E,M,1h} = 0.55$). That the mesoscale vertical velocity variance is sensitive to the averaging scale is in line with analyses in Stephan and Mariaccia (2021), who used radiosondes from the EUREC⁴A ship array to show that the divergence amplitudes at equivalent radii of 100–300 km scale approximately inversely with radius (as in ERA5 and ICON, which is consistent with results of Bony and Stevens, 2019). They also show that the Radius⁻¹ scaling of divergence amplitudes is valid up to synoptic scales of about 1000 km. We also find the expected pronounced sensitivity of cloud-base W to the averaging scale in ERA5 (see Fig. 4 left).

We also check the scale sensitivity of the surface buoyancy flux, which contributes most to variability in E (ED Fig. 5b). Despite its overestimation compared to the dropsonde-derived flux, the SBF in ERA5 captures the observed variability well ($r=0.81$, see discussion in L437ff) and can thus be useful to study the scale sensitivity. The SBF in ERA5 shows only little scale dependency (Fig. 4 right). This is likely because variability in the SBF is mostly controlled by the surface wind speed (see Fig. 6d below) and radiative cooling (Naumann et al. 2019), both of which are large-scale. The surface wind speed has autocorrelation coefficients of 0.74 after two days and 0.48 after 8 days (Fig. 3d of Albright et al. 2022), highlighting the larger-scale control of the wind speed. On the other hand, the surface buoyancy flux shows a pronounced diurnal cycle, driven by the diurnality of the surface wind (Savazzi et al. 2022, Vogel et al. 2020). For much longer temporal averaging, SBF variability would therefore also decrease. The other term that contributes to variability in E, $\Delta\theta_v$, is not expected to become less variable at larger scales if one takes care to average in ways that preserves the local gradients.

Fig. 4: Standard deviation of 925 hPa vertical velocity (W) and surface buoyancy flux (SBF) from hourly ERA5 output from 20 Jan – 20 Feb 2020 using different domain sizes (from $1^\circ \times 1^\circ$ to $8^\circ \times 8^\circ$) centered on the EUREC⁴A circle (but excluding Barbados). The SBF is scaled by 0.93 K^{-1} (such that it fits the mean observed entrainment rate E) to have roughly comparable orders of magnitude for the sd. Note that the absolute values cannot be compared here.

The temporal fluctuation and horizontal advection of h (additional terms in eq. 1 that are excluded in the equilibrium M focused on) are computed only on the 3h timescale, as they tend to be noisy if only a small number of sondes are used to compute them, and as the circular flight path of HALO confounds both spatial and temporal variability within one circle. In ICON-LES (Vogel et al. 2020), we found that the advection term tends to be larger on smaller domains, both due to h being noisy and local gradients being quite large if computed over small areas.

Regarding the GCMs: Indeed, the GCMs don't represent mesoscale motions, but have a 'large-scale' vertical velocity at the grid-scale. The models have horizontal grid spacings of $\sim 1.4^\circ$ - 2.2° , so the grid-scale vertical velocity should be somewhat comparable in scale to the EUREC⁴A circle observations. Nevertheless, the models don't represent the (sub-grid) processes likely leading to the observed variability in the mesoscale vertical velocity (e.g. shallow circulations driven by mesoscale moisture anomalies, radiatively-driven circulations, or local SST gradients). We therefore don't expect the current generation of models to represent the observed spectrum

of W variability. Our results therefore demonstrate the importance of better understanding the processes that lead to the variability in mesoscale vertical velocity, and to find ways to reproduce some of this variability in climate models.

In conclusion: the mass budget terms are scale sensitive and M variability will decrease on larger scales. W is more sensitive to scale than E , such that the contribution of W to M variability would likely become smaller compared to the contribution of E on scales larger than about $4^\circ \times 4^\circ$. However, also the variability in E reduces at larger scales, especially on timescales longer than a day.

- Changes applied in the revised manuscript

Apart from clarifying the scale definition at the beginning of the Methods section and renaming circling-/circle-scales to 3h-/1h-scales, we applied the following changes in the revised manuscript:

L177f | Main text:

[All the models also strongly underestimate variability in W (Extended Data Fig. 9b)], as they do not represent the sub-grid processes leading to the observed variability in the mesoscale vertical velocity (e.g., shallow circulations driven by differential radiative cooling (Naumann et al. 2017) or local SST gradients (Lambaerts et al. 2013)).

L356ff | Methods subsection 'EUREC⁴A campaign':

The spatial scale of the observations represents the lower end of Orlanski 1975's meso- α scale and is comparable in size to a climate model grid box. The 200-300km scale is the relevant scale of the cloud processes for a trade cumulus ensemble and also the scale that convective parameterizations target. It lies in between the $O(1 \text{ km})$ scale of individual clouds and the synoptic scale of $O(1000\text{km})$, and is associated with the emergence of the four prominent cloud organization patterns of Stevens et al. 2019. As the airmasses are advected by about 30km per hour (at the campaign-mean wind speed of about 9ms^{-1} at 1km height), the spatial sampling of the 220 km diameter circle does not differ substantially between the 1h and 3h timescales, which motivates our nomenclature focus on the time rather than space scale. Using the measurements, model and reanalysis data we would not expect our results to change substantially if the analysis domain were increased or reduced by a factor of two or more (see Methods subsection 'Mass flux estimation' for a discussion of the scale sensitivity of the results).

L517-535 | Methods subsection 'Mass flux estimation'

The mass budget terms show different degrees of scale sensitivity (see also discussion in Vogel et al. 2020). ED Fig. 3c and 5a show that the correlation between W and M is slightly larger at the 1h-scale compared to the 3h-scale ($r_{W,M,3h} = 0.60$ and $r_{W,M,1h} = 0.67$), while they are essentially the same for E and M ($r_{E,M,3h} = 0.54$ and $r_{E,M,1h} = 0.55$). The scale sensitivity of the W variance is in line with radiosonde data from the EUREC⁴A ship array, which show that the divergence amplitudes at equivalent radii of 100–300 km scale inversely with radius (Stephan and Mariaccia, 2020) (as in ERA5 and ICON, consistent with Bony and Stevens, 2019). In ERA5, the scale sensitivity of the surface buoyancy flux, which contributes most to variability in E (ED Fig. 5b), is much smaller compared to the scale sensitivity of W (not shown). This is likely because variability in the SBF is mostly controlled by the surface wind speed (ED Fig. 5h) and radiative cooling (Naumann et al. 2019), both of which are large-scale. The surface wind speed has autocorrelation coefficients of 0.74 for a two day and 0.48 for an eight day lag (Fig. 3d of Albright et al. 2022). Instead, the diurnality in the surface wind (Savazzi et al. 2022) causes a pronounced diurnal cycle of the surface buoyancy flux (Vogel et al. 2020) and thus a sensitivity of E to much longer temporal averaging. Also, the variability in the temporal fluctuation and horizontal advection of h (eq. 1) decreases on larger scales (Vogel et al. 2020). In summary, M variability decreases on larger averaging scales. The scale sensitivity of W is larger compared to E , such that the contribution of W to M variability tends to become smaller compared to the contribution of E on much larger scales.

2) There are some minor issues in the uncertainty discussion. There is an overemphasis in this paper on the first measurements of key variables and relationships. It feels to me that those are estimates based on observations, not direct measurements. For example, the mass flux M is based on the residuals of the budget equation and the entrainment E is based on a fixed portion of the surface flux. Cloudiness C from aircraft remote sensors is based on a novel approach but has inherent uncertainties (e.g., Cloudiness near cloud base of shallow cumulus clouds can be highly dependent on the height, and small changes in flight level can make a large difference; Droplets are often too small to be observed by radar).

We agree that our mass flux estimates are not direct measurements but estimates based on mixed-layer framework applied to observations. EUREC⁴A represents a step-change in our ability to close mixed layer budgets with observations. For the first time, we have such a number of repeated observations of W , such an intensive sampling of thermodynamic profiles, and such a large set of surface flux estimates to constrain the budgets in unprecedented detail. We can thus derive all the terms with observations and do not need reanalyzes for it (e.g., as used for W in Ghate et al. 2019). Further, while ground-based doppler radars allow to derive a cloud mass flux for non-precipitating clouds (Ghate et al. 2011, Lamer et al. 2015, Klingebiel et al. 2021), the single-point perspective yields either mass fluxes for individual cloud objects or ensemble averages over a long period (~20h averaging time needed to get a representative large-scale cloud fraction as calculated in Klingebiel et al. 2021). Our mass budget derived observational estimate of M is at the mesoscale, which is the relevant scale of the studied cloud processes (see reply above).

We clarified the difference between observation and direct measurement of M by replacing 'measurement(s)' with 'observation(s)' of M throughout the manuscript. We also changed the first subheading to '*Observations of M , C and R co-variations*' (instead of 'First measurements of M , C , and R co-variations').

Regarding the uncertainties of the cloud-base cloud fraction C , Fig. 18 of Bony et al. 2022 shows that despite such uncertainties, the magnitude and variability of independent C estimates from the different instruments onboard ATR agree very well. As our default C , we use the radar-lidar synergy product BASTALIAS, which provides a best estimate in terms of footprint and sensitivity. See our discussion in L386-398 and L706-734.

We agree that the cloud fraction varies systematically with height, and missing the correct level with ATR could lead to an underestimation of C . We used a range of different instruments to evaluate the sub-cloud layer height definition for every 3h-circles set (see Fig. 3 below for an example). From this analysis, we deduced that we usually flew at a level close to the 'true' sub-cloud layer top as computed from the dropsondes. We also used the BCO cloud radar to see how much the variability and magnitude of C changes when the level of maximum cloud-base cloud fraction is missed by ± 60 m. Only if our measurements deviated from the height of maximum cloudiness in ways that co-varied with M would we expect such a bias to influence our analysis.

We summarized this analysis in the last paragraph of the Methods subsection 'Observations' (now L407-420), and added the sentence in bold:

*"The BCO cloud radar data also demonstrates that missing the level of maximum cloud-base cloud fraction in 3-h averages by e.g. 60 m does not affect the variability of C (correlations of $r=0.99$ and $r=0.93$ with the maximum C when 60 m above and below the peak level, respectively), and only marginally affects its magnitude (18% and 33% smaller relative to the maximum C for being 60 m above or below the peak level, respectively). **So only if the ATR flight level deviated from the height of maximum cloudiness in ways that co-varied with M would we expect such a height difference to influence our analysis.** As the ATR aircraft*

usually flew a bit above h (Extended Data Fig. 4a), and because it sampled much more clouds in 3 h compared to the stationary BCO, a potential influence of missing the peak level is deemed not to bias our findings.”

Fig. 5: Vertical profiles of (a) θ_v and (b) different variables to estimate the sub-cloud layer top for the first 3h-circles set on 11 February. In (a), the profiles are shown for the 3 individual 1h-circles (solid excludes cold-pool soundings, dotted includes them), and the dashed horizontal lines show the respective sub-cloud layer height h . Shown in (b) are the radar-derived cloud fraction profiles (BCO CF and Meteor CF), pdfs of ceilometer first-detected cloud base heights (BCO cbh and Meteor cbh) for both the BCO and R/V Meteor, as well as a pdf of sub-cloud layer top estimates from the WALES lidar onboard HALO. The horizontal lines are the dropsonde-derived h (black) and the mean ATR flight level (red). To ease comparison, the data are scaled by their maximum value (i.e., x/x_{max}).

Referee #3 (Remarks to the Author):

I have reviewed "Strong cloud-circulation coupling explains weak trade cumulus feedback" by Vogel et al. The authors present an analysis of covariability of convective mass flux, large-scale vertical velocity, entrainment, relative humidity, and cloud fraction in the tradewind cumulus cloud regime. The two main conclusions are that observational results are inconsistent with the mixing-desiccation hypothesis, reducing the plausibility of a strongly positive cloud feedback in tradewind cumulus. A somewhat related conclusion is that GCMs overestimate the sensitivity of cloud fraction to thermodynamic (RH) changes and underestimate its sensitivity to circulation.

These are highly relevant and timely conclusions of interest to a broad range of scientists, not least because they contribute to the intense discussion of potentially high climate sensitivity in the wake of CMIP6. As part of this discussion, a great deal of process realism analysis has been done on midlatitude clouds (e.g., Tan et al. 2016, McCoy et al. 2020, Mulmenstadt et al. 2021, Fraser et al. 2022), but the equally or perhaps even more important tropical shallow cumulus regime (Zelinka et al. 2020, Myers et al. 2021) has received much less process attention. It is refreshing to see this well designed study address this important cloud type.

As far as I can tell, the analysis is sound. My main concern is a nagging doubt about using present-day (monthly) variability to make a direct statement about a very low-frequency perturbation like anthropogenic warming. I would feel better if we had a modeling system that allowed us to make the link across time scales. As the authors demonstrate, the models we do have are all quite far removed from process realism, so I am not sure how far dividing them into "very unrealistic" and "even more unrealistic" can get us in terms of estimating what is realistic. This is certainly not a criticism of this study, more a criticism of the tools we currently have available to us. Or perhaps it is a lengthy way of saying that the part of the paper I found of greatest relevance was the refutation of the mixing-desiccation storyline, not the reading of the CMIP tea leaves (Fig. 3c).

We agree that the most relevant contribution of our work is testing (and eventually refuting) the mixing-desiccation hypothesis with observations. Testing what the CFMIP models do was a secondary objective. Before EUREC4A, we didn't have observations of the magnitude, variability, and relationship of M, C and R at scales comparable to GCMs. So testing the mixing-desiccation idea with data advances our understanding and may thus also guide future model development.

That said, we do not trust any of the models to provide a realistic feedback estimate. We therefore neither say that a specific feedback strength is the most plausible, nor do we try to explicitly provide a constraint on a realistic feedback strength (e.g., through using the correlation of M and C as an emergent constraint, see Fig. 2 and the corresponding reply to Referee #1). What we can say is that the models predicting a strong positive feedback are unrealistic. Yet, a number of models predicting a weak feedback are also unrealistic. A model could be assessed to be realistic in its trade cumulus feedback when it will at least reproduce the C~M and C~RH relationships in agreement with observations.

To clarify this point more in the manuscript, we emphasize again in the end that testing the idea is important. We do this by adding another subheading before the CFMIP model section ("*Models underestimate strong cloud-convection coupling*") and moving the '*Implications for trade cumulus feedbacks*' subheading before the third from last paragraph. This helps to emphasize that the implications for the trade cumulus feedback strength are not solely due to model analysis, but mostly due to refuting the mixing-desiccation mechanism, a mechanism that would lead to a strong feedback.

The comments below are offered as constructive criticism. If they are easy to implement, I think they may assist the reader; if they are time-consuming, then I would rather that the authors

ignore these suggestions than that they hold up publication.

* My main substantive question concerns the entrainment calculation that enters into the calculation of M . The entrainment estimate is a parameterization based on more readily observable quantities and, as such, relies on parameter estimation. Did the authors account for the possibility that the parameter estimate itself could depend on the mesoscale environment? I ask because such a parameter dependence could spuriously produce or suppress covariabilities between M and W , C , or R . Since the authors' conclusions rely on covariability between C and $\{W, R, M\}$, spurious covariability introduced by the entrainment parameterization would be problematic.

In our mass budget formulation, we parameterize E using a modified version of the flux-jump model, which accounts for the $\sim 150\text{m}$ thick transition layer between the mixed-layer and sub-cloud layer tops. The uncertain parameters A_e and weighting factors C_θ and C_q are estimated through a joint Bayesian inversion to close the heat and moisture budget by Albright et al. (2022). So the parameter estimates are a priori independent of M and W . Also, the synoptic variability during EUREC⁴A can be well explained by keeping these parameters constant. Albright et al. (2022) also explored to what extent other factors (like wind speed or wind shear) correlated with residuals in their Bayesian fits and found no evidence of a systematic effect of other factors.

Fig. 6: Relationships of (a) E , (b) SBF , and (c) θ_v -jump versus W , and relationships of 10m wind speed versus (d) SBF , (e) E and (f) W , shown for both the 3h-scale (filled) and 1h-scale (open circles, with the corresponding correlation coefficient denoted as $r.c.$).

The surface buoyancy flux (SBF) and the θ_v -jump across h , however, might depend more on the mesoscale environment than the constant parameters. We find that E and W are anticorrelated with $r = -0.35$ on the 3h-timescale (Fig. 6a), which is mostly due to an anticorrelation of W with the surface buoyancy flux. One important control of trade-wind cloudiness found in previous studies is the surface wind speed (Nuijens et al. 2015, Brueck et al. 2015). The wind speed could thus be a physical mechanism through which W and SBF might be connected. The surface wind speed explains 40% of variability in the SBF (Fig. 6d), but is hardly correlated with W ($r = -0.13$, Fig. 6f). The surface wind speed is a larger-scale signal and set mostly by the large-

scale pressure gradient, with autocorrelation coefficients of 0.74 after two days and 0.48 after 8 days (Fig. 3d of Albright et al. 2022). Variability in the SBF in ERA5 shows only little dependency on the averaging scale (see Fig. 4b and its discussion above). The SBF and thus also E (as the SBF explains most variability in E) are therefore mostly controlled on scales larger than the ~200km scale of mesoscale motions.

Also, the fact that E and W (and SBF and W) are anticorrelated (Fig. 6a-b) means that they do not spuriously compensate for each other. So both statistically (from the anticorrelation) and physically through the scale argument (larger-scale wind signal vs. mesoscale motions in W), we believe that our parameterization of E does not induce structural dependence that would spuriously produce a covariability between W & M and C.

To clarify these issues in the manuscript, we added Fig. 6a&d to Extended Data Fig. 5 as additional panels. Furthermore, we added the following paragraph to the Methods subsection 'Mass flux estimation' (L536ff):

“As noted above, E describes the net effect of local processes and must be inferred from the statistics of other quantities (i.e., the mean sub-cloud layer growth rate, or the dilution of sub-cloud layer properties). This raises the question if the E estimate itself might depend on the mesoscale environment and therefore introduce spurious co-variabilities between M, W and C. The Bayesian estimation of the uncertain parameter estimates A_e , C_q and C_θ is a priori independent of M and W. Also, the synoptic variability during EUREC4A can be well explained by keeping them constant (Albright et al. 2022). Albright et al. (2022) also explored to what extent other factors correlated with residuals in their Bayesian fits and found no evidence of a systematic effect of other factors, including windspeed and shear (Canut et al. 2012). As discussed above, the variability in E tends to be less scale-sensitive than W, and mostly controlled by larger-scale factors like the surface wind speed (through the surface buoyancy flux, Extended Data Fig. 5b,h). Furthermore, E and W are anticorrelated ($r_{E,W} = -0.35$, Extended Data Fig. 5g). So both statistically from the anticorrelation and physically through the scale argument, we believe that our parameterization of E does not induce spurious co-variability.”

* I was slightly perplexed that the analysis is presented as the *raison d'être* of the EUREC4A campaign and synthesizes many of the data streams collected during the campaign, but that the author list and acknowledgments do not reflect the many scientists and engineers who contributed to data collection. I suppose citing the campaign description papers is an alternate form of acknowledgment, and maybe this is a cultural difference.

The analysis presented in this paper is the original motivation for initiating the EUREC4A field campaign as presented in Bony et al. 2017 and motivated the particular flight strategy that is exploited here. During the preparation of the campaign, many more teams and platforms joined and the scope of the campaign was greatly expanded. Regarding authorship, the philosophy was to “recognize all technical/scientific contributions to the data collection” in the overview paper of Stevens et al. 2021. Beyond the overview paper, authorship is more restrictive, such that on specific data papers only those closely involved are included as co-authors. The same is true for scientific papers. This means that for the present paper, all authors have either made substantial contributions to the data analysis or the conception and design of the work. This is in line with a common consensus of good scientific practice in our field, which tends to define authorship rather restrictively (e.g. as formulated in MPI-M's Good Scientific Practice policy¹).

In the revised manuscript, we recognize in the acknowledgments the help of a larger number of colleagues who shared additional data for comparison or gave helpful advice. We also included a more general statement on funding support for EUREC4A. Furthermore, the Data Availability Statement now explicitly mentions all the individual datasets used.

¹ https://mpimet.mpg.de/fileadmin/publikationen/Volltexte_diverse/Good_Scientific_Practice.pdf

* The authors describe the analysis as a "process-based constraint". However, I could not find an explicit statement of which process(es) the authors claim to be constraining. Entrainment? Organization? Convection? The "convection" process in GCMs? I am not disagreeing about the label, but I think more specificity would avoid the buzzwordy taste that "process-oriented" is starting to acquire.

We constrain the strength of the cloud-convection coupling with the EUREC⁴A data, at the relevant scale that convective parameterizations target. We agree that both words, process-based and constraint, are a bit overused. So we removed them in the revised manuscript, e.g.:

- In the Summary, we now write *'Our observational analyses render models with large positive feedbacks implausible, ...'* (instead of *'The process-based constraints render models with large positive feedbacks implausible, ...'*)

- in L200-202, we write *"The EUREC⁴A data thus provide a physical test of the capacity of models to represent the interplay of the processes active in regulating trade-wind cloud amount, and may guide future model development."* (instead of *'The process-based constraints presented here can thus provide guidance to future model development.'*)

- We also modified the sentence in L214-216: *The EUREC⁴A observations of the physical processes that drive the short-term variability of C thus rule out the mechanism that leads to the largest positive trade cumulus feedbacks in current climate models.* (instead of *'The EUREC⁴A constraints on the processes that drive the short-term variability of C thus rule out the physical mechanism that leads to the largest positive trade cumulus feedbacks in current climate models.'*)

* On the presentation, I found the use of "circle" and "circling" to mean different things quite confusing.

(Same as reply to Ref. #1, Major comment #2:)

We used this circle vs. circling nomenclature as it was introduced in the EUREC⁴A overview paper (Stevens et al. 2021) and has become standard in the following (e.g. Konow et al. 2021, Albright et al. 2022). Nevertheless, we agree that it can be confusing and thus completely drop the 'circle / circling' nomenclature and only keep the timescale prefixes. So the (3h-)circling scale becomes the 3h-scale, and the (1h-)circle scale becomes the 1h-scale. When referring to a specific 3h "circling" period, we refer to it as a '3h-circle set(s)'.

As the airmasses are advected by ~30km per hour (at the campaign-mean windspeed of ~9 m/s at 1km height), the spatial sampling of the ~220km diameter circle doesn't differ substantially between the 1h and 3h timescales, such that it makes sense to focus the nomenclature on the time rather than the space scale.

* Further on the presentation, I don't love that M is frequently referred to as "mass flux" when its units are velocity.

It is quite common to specify the mass flux in velocity units, as the density doesn't vary much and is often assumed to be constant (e.g. Ghate et al. 2011, Klingebiel et al. 2021). But to clarify this early on (and not just at the end of the 'Mass flux estimation' Methods subsection), we added in connection to eq. 1 (L67):

"Note that we define M as the (mass) specific mass flux, which has units of velocity (see Methods)."

We also modified the description in the Methods subsection 'Mass flux estimation' (L507-511): *"Also note that our M is defined as the (mass) specific mass flux and has units of velocity. It differs from the more familiar mass flux (in units of $\text{kg m}^{-2} \text{s}^{-1}$) by the air density, which is usually assumed to be constant (Ghate et al. 2011, Klingebiel et al. 2021), and which is justified here given the small variation of h (Extended Data Fig. 4a) and surface pressure (less than 0.2% of the mean) across the measurements."*

* l. 36: "make up for"  "make up" **Changed**

References

- Albright, A. L., S. Bony, B. Stevens, and R. Vogel (2022), Observed subcloud layer moisture and heat budgets in the trades, *J. Atmos. Sci.*, doi:10.1175/JAS-D-21-0337.1.
- Bigorre, S. P., and A. J. Plueddemann (2021), The annual cycle of air-sea fluxes in the northwest tropical atlantic, *Frontiers in Marine Science*, 7, doi:10.3389/fmars.2020.612842.
- Bony, S., and B. Stevens (2019), Measuring area-averaged vertical motions with dropsondes, *J. Atmos. Sci.*, 76(3), 767–783, doi:10.1175/JAS-D-18-0141.1.
- Bony, S., B. Stevens, F. Ament, et al. (2017), Eurec4a: A field campaign to elucidate the couplings between clouds, convection and circulation, *Surv. Geophys.*, 38, 1529, doi:10.1007/s10712-017-9428-0.
- Bony, S., M. Lothon, J. Delanoë, et al. (2022), Eurec⁴a observations from the safire atr42 aircraft, *Earth Syst. Sci. Data*, 14(4), 2021–2064, doi:10.5194/essd-14-2021-2022.
- Brueck, M., L. Nuijens, and B. Stevens (2015), On the seasonal and synoptic time-scale variability of the north atlantic trade wind region and its low-level clouds, *J. Atmos. Sci.*, 72(4), 1428–1446, doi:10.1175/JAS-D-14-0054.1.
- Caldwell, P. M., M. D. Zelinka, and S. A. Klein (2018), Evaluating emergent constraints on equilibrium climate sensitivity, *J. Climate*, 31(10), 3921–3942.
- Canut, G., F. Couvreux, M. Lothon, D. Pino, and F. Saïd (2012), Observations and large-eddy simulations of entrainment in the sheared sahelian boundary layer, *Boundary-Layer Meteorology*, 142(1), 79–101, doi:10.1007/s10546-011-9661-x.
- Ghate, V. P., M. A. Miller, and L. DiPreto (2011), Vertical velocity structure of marine boundary layer trade wind cumulus clouds, *J. Geophys. Res. Atmos.*, 116(D16), doi:10.1029/2010JD015344.
- Ghate, V. P., D. B. Mechem, M. P. Cadeddu, et al. (2019), Estimates of entrainment in closed cellular marine stratocumulus clouds from the magic field campaign, *Quart. J. Roy. Meteor. Soc.*, 0(ja), doi:10.1002/qj.3514.
- Klingebiel, M., H. Konow, and B. Stevens (2021), Measuring shallow convective mass flux profiles in the trade wind region, *J. Atmos. Sci.*, 78(10), 3205 – 3214, doi:10.1175/JAS-D-20-0347.1.
- Konow, H., F. Ewald, G. George, et al. (2021), Eurec⁴a's HALO, *Earth Syst. Sci. Data*, 13(12), 5545–5563, doi:10.5194/essd-13-5545-2021.
- Lamer, K., P. Kollias, and L. Nuijens (2015), Observations of the variability of shallow trade wind cumulus cloudiness and mass flux, *J. Geophys. Res. Atmos.*, 120(12), 6161–6178, doi:10.1002/2014JD022950.
- Medeiros, B., and L. Nuijens (2016), Clouds at barbados are representative of clouds across the trade wind regions in observations and climate models, *Proc. Natl. Acad. Sci. U.S.A.*, 113(22), E3062–E3070, doi:10.1073/pnas.1521494113.
- Myers, T. A., R. C. Scott, M. D. Zelinka, S. A. Klein, J. R. Norris, and P. M. Caldwell (2021), Observational constraints on low cloud feedback reduce uncertainty of climate sensitivity, *Nature Climate Change*, 11(6), 501–507, doi:10.1038/s41558-021-01039-0.

Nuijens, L., B. Medeiros, I. Sandu, and M. Ahlgrimm (2015), Observed and modeled patterns of covariability between low-level cloudiness and the structure of the trade-wind layer, *J. Adv. Model. Earth Syst.*, 7(4), 1741–1764, doi:10.1002/2015MS000483.

Naumann, A. K., B. Stevens, C. Hohenegger, and J. P. Mellado (2017), A conceptual model of a shallow circulation induced by prescribed low-level radiative cooling, *J. Atmos. Sci.*, 74(10), 3129 – 3144, doi:10.1175/JAS-D-17-0030.1.

Naumann, A. K., B. Stevens, and C. Hohenegger (2019), A moist conceptual model for the boundary layer structure and radiatively driven shallow circulations in the trades, *J. Atmos. Sci.*, 76(5), 1289 – 1306, doi:10.1175/JAS-D-18-0226.1.

Orlanski, I. (1975), A rational subdivision of scales for atmospheric processes, *Bull. Amer. Meteor. Soc.*, 56(5), 527–530.

Savazzi, A. C. M., L. Nuijens, I. Sandu, G. George, and P. Bechtold (2022), The representation of winds in the lower troposphere in ecmwf forecasts and reanalyses during the eurec4a field campaign, *Atmos. Chem. Phys. Disc.*, 2022, 1–29, doi:10.5194/acp-2021-1050.

Schlund, M., A. Lauer, P. Gentine, S. C. Sherwood, and V. Eyring (2020), Emergent constraints on equilibrium climate sensitivity in cmip5: do they hold for cmip6?, 11(4), 1233–1258, doi:10.5194/esd-11-1233-2020.

Stephan, C. C., and A. Mariaccia (2021), The signature of the tropospheric gravity wave background in observed mesoscale motion, *Weather Clim. Dynam.*, 2 (2), 359–372, doi:10.5194/wcd-2-359-2021.

Stevens, B., S. Bony, H. Brogniez, et al. (2020), Sugar, gravel, fish and flowers: Mesoscale cloud patterns in the trade winds, *Quart. J. Roy. Meteor. Soc.*, 146(726), 141–152, doi:https://doi.org/10.1002/qj.3662.

Stevens, B., S. Bony, D. Farrell, et al. (2021), Eurec⁴a, *Earth Syst. Sci. Data*, 13(8), 4067–4119, doi:10.5194/essd-13-4067-2021.

Vogel, R., S. Bony, and B. Stevens (2020), Estimating the shallow convective mass flux from the subcloud-layer mass budget, *J. Atmos. Sci.*, 77(5), 1559 – 1574, doi:10.1175/JAS-D-19-0135.1.

Reviewer Reports on the First Revision:

Referees' comments:

Referee #1 (Remarks to the Author):

Thank you for your replies and updates. The main manuscript text is very clear and the findings are interesting and highly relevant. In this review, as requested by the editor, special attention was paid to the methods section and the figures/captions.

Further questions from first review:

- How do I reconcile L164-165 with your statement in the methods (L616) and response to previous comments that the models are not erroneously producing stratocumulus clouds in the EUREC4A domain?
- I understand your argument for not doing an emergent constraint analysis given that almost none of the models seem to get the right answer for the right reasons since they lack mesoscale structure. I would then just caution against making such bold statements like L217-218 “our findings refute an important line of evidence for a large climate sensitivity” and keep the discussion more tightly focused on the goal of refuting the mixing-desiccation mechanism, which I think is quite robust and well demonstrated.

Comments on methods:

- L355: “excluded from the calculated correlations”
- L375: rephrase the parenthetical to make clear that ERA5 and ERA-Interim both are significantly drier than the dropsonde-measured humidity. “offset” is not clear about if the measurements were larger or smaller than the reanalysis.
- L519: typo. Both correlation coefficients are labeled as “3h”
- L528-531: I don’t understand these two sentences. Are you saying the surface wind variations have a long multi-day timescale or a diurnal timescale? Please clarify. The use of the word “instead” is confusing here.
- L553: How meaningful is it to compute a standard deviation with only 3 data points? There is no way here to confirm the assumption of Gaussianity.
- L557: Is there also measurement uncertainty for each estimate of C that you add in quadrature? Or if not, why do you ignore this?
- L590: Do you also assume here density is constant? Or if you have a measurement of density, why do you say you assume it constant in L509?
- L620: 0.25% seems very small. What is the argument to include these profiles at all rather than impose a threshold for minimum cloudiness to consider from the GCMs?
- L641: You use the same regime definitions for amip and amip4K, right? I assume you chose the grid points to compare based on if this trade Cu definition held in the base simulations (amip) and then look at the same grid points in amip4K. Please clarify.

Comments on figures/captions:

- Fig 1.

- o What are the “X” symbols in panel d?
 - o Do the relative humidity measurements also have observational uncertainty? Can that be added to panel d?
- Fig 2.
 - o Can error bars be added to panel a) on the reconstructed C values?
 - o Why are there no error bars shown on panel b) when these data are already shown in c) and d)?
 - o It would be nicer to have the r value in the top left of every panel, including b).
- Fig 3.
 - o It might be better to color the models by their feedback strength (not binned) and then show the color bar explicitly?
 - o Panel c) What do you mean by “frequent stratocumulus”? Can you be quantitative here.
 - o I’m not sure how useful it is to modify the extreme outlier models to fit onto your axes. Maybe it is better to just leave them off and note the values in the caption? Or maybe it is better to increase the axes to accommodate all models? The way you have it on panels a) and c) is hard to parse because naturally our eyes will try and find trends between the models which are misrepresented here.
 - o Panel c) – It seems like the observational uncertainty bars are not shown on the plot for the high end of aM/aR. Please fix.
- ED Fig 1. I still really like this figure. Is there space to add it to the main text?
- ED Fig 2.
 - o Panel a) should have M’ not M` in the legend. Check this in other figures too.
 - o Panel b) please update the terminology of “circling” vs “circle” to be consistent with the changes to “1h” and “3h”
 - o Can you change “r” and “r.c” to “r_{1h}” and “r_{3h}” for the correlation coefficients?
 - o It would be nice to show the error bars on these scatter plots as you have in Fig 2
- ED Fig 3. What are the dashed lines on panels b) and d)?
- ED. Fig 4. Like Fig 1., what are the “X” symbols?
- ED. Fig 5.
 - o Change legend “circle and circling”
 - o What is “IU” on panel h)? Can you just write out “10-m WS” or something more descriptive?
 - o What are the grey symbols here? Please restate this definition in the figure caption.
 - o What are the “x” vs “o” markers indicating? Please add this to the caption.
 - o Note in caption what the dotted lines are in panels a) and g). 1:1, yes?
- ED. Fig. 6.
 - o Correct M` to M’
 - o Make caption text describing a) and b) parallel.
- ED. Fig. 7. Elsewhere cloud cover is written just as “C” not “CC”

- ED. Fig. 9.
 - o Change 1:1 lines to dotted to match other figures and include this in the caption.
 - o Indicate that the colors of the markers matches previous figures and they are colored by feedback strength.
 - o Are “stratocumulus-like conditions” just when $R > 94\%$? Or is there another criteria? This is unclear to me from the text and the caption.
 - o “standard deviation σ of C”  “standard deviation of C (σ_C)”

Referee #2 (Remarks to the Author):

The authors addressed my previous comments. I appreciate the time and efforts.

Referee #3 (Remarks to the Author):

I have reviewed the revised manuscript by Vogel et al. My comments on the initial manuscript were fairly minor, and I see that the authors have addressed all of them to my satisfaction.

At the request of the editor, I have taken a closer look at the statistical methods and uncertainty treatments. I may have already remarked in the initial review that I was happy to see the statistical methods were very clearly described, and this has not changed. The meaning of error bars in the figures is defined in the captions, either directly or by reference to the methods section. The statistics themselves appear sound to me. Some of the quoted uncertainties are a combination of statistical (i.e., sampling) and systematic, but that's fine; it would be overkill to quote them separately. Most of the systematic uncertainties are handled via the (absence of) sensitivity of the results to parameter choices in a "robustness" section of the Methods; that's also fine, and I commend the authors on the extensive list of sensitivities they have investigated. None of the statistical methods employed in the analysis struck me as inappropriate or misleading. I have compared a handful of the error bars in the manuscript to back-of-the-envelope calculations of individual error bars and did not find anything astonishing.

Author Rebuttals to First Revision:

Response to referees

Nature manuscript 2022-05-07332A entitled "Strong cloud-circulation coupling explains weak trade cumulus feedback" by Vogel et al. 2022.

We thank the editor and the three referees for their feedback on the revised manuscript. Referee 1 raises a few additional points that we are happy to address in our revision. These points are addressed below, with our responses inserted in green.

Referee #1 (Remarks to the Author):

Thank you for your replies and updates. The main manuscript text is very clear and the findings are interesting and highly relevant. In this review, as requested by the editor, special attention was paid to the methods section and the figures/captions.

We thank the reviewer very much for going through the methods and figures in such detail. We really appreciate the effort.

Further questions from first review:

- How do I reconcile L164-165 with your statement in the methods (L616) and response to previous comments that the models are not erroneously producing stratocumulus clouds in the EUREC4A domain?

In terms of the large-scale environment (i.e. the 700 hPa pressure velocity and the estimated inversion strength), the cfSites locations of the models fall into the trade cumulus regime. So the models are getting the location of the trade cumulus regime right. However, the models erroneously produce stratocumulus under these climatological trade cumulus conditions. This is due to the models having unrealistic physics in the trade cumulus regime. To clarify this, we rephrased the sentence in L616 as follows:

*"We verified that **in terms of the large-scale environment**, the cfSites used fall into the **climatological trade cumulus regime** as defined by Myers et al. (2021)."*

(instead of: *"We verified that the cfSites used fall into the trade cumulus regime as defined by Myers et al. (2021)."*)

- I understand your argument for not doing an emergent constraint analysis given that almost none of the models seem to get the right answer for the right reasons since they lack mesoscale structure. I would then just caution against making such bold statements like L217-218 "our findings refute an important line of evidence for a large climate sensitivity" and keep the discussion more tightly focused on the goal of refuting the mixing- desiccation mechanism, which I think is quite robust and well demonstrated.

We believe it is exactly by refuting the mixing-desiccation mechanism that we refute an important line of evidence for a strong trade cumulus feedback. The understanding of the interaction between convective mixing and cloudiness gained here allows us to refute a proposed mechanism for a large positive trade cumulus feedback, and thus a physical storyline (or hypothesis) for a large climate sensitivity (Stevens et al. 2016). The fact that models representing the mixing-desiccation mechanism have particularly strong feedbacks further supports our argument, but it is not necessary.

To make that more explicit, we moved the sentence in L217-218 to the last paragraph of the main text (rather than the end of the 2nd-from last paragraph that still discussed the climate models), and slightly rephrased it (in bold). The last paragraph now starts as follows:

“By showing that mesoscale motions inhibit the mixing-desiccation mechanism, we refute an important physical hypothesis for a large trade cumulus feedback. In the spirit of the *story-line* approach for constraining equilibrium climate sensitivity¹⁰, our findings thus **refute an important line of evidence for a strong positive cloud feedback and thus a large climate sensitivity.**”

Comments on methods:

- L355: “excluded from the calculated correlations” Changed
- L375: rephrase the parenthetical to make clear that ERA5 and ERA-Interim both are significantly drier than the dropsonde-measured humidity. “offset” is not clear about if the measurements were larger or smaller than the reanalysis. Changed to: “...albeit with a 10% larger 850hPa relative humidity during EUREC⁴A **(the EUREC⁴A dropsondes also have an ~8% larger relative humidity compared to the 2013-2022 average in ERA5, not shown).**”
- L519: typo. Both correlation coefficients are labeled as “3h” Changed. Thanks for spotting this!
- L528-531: I don’t understand these two sentences. Are you saying the surface wind variations have a long multi-day timescale or a diurnal timescale? Please clarify. The use of the word “instead” is confusing here. We removed the word ‘instead’ and clarified the sentence about the diurnality of the wind as follows:

“The surface wind speed has autocorrelation coefficients of 0.74 for a two day and 0.48 for an eight day lag (Fig. 3d of Albright et al. 2022). **Although weaker compared to the synoptic variability, the surface wind also has a distinct diurnal cycle (Vial et al. 2021, Savazzi et al. 2022), which causes a diurnal cycle of the surface buoyancy flux (Extended Data Fig. 1c and Vogel et al. 2020b). Some of the diurnal variability in E is thus lost for longer temporal averaging.**”

- L553: How meaningful is it to compute a standard deviation with only 3 data points? There is no way here to confirm the assumption of Gaussianity. Each of the 3 data points represents about 12 individual dropsondes (for R and M) or 50min of flight (for C), such that one point effectively represents more data. Furthermore, it is not necessary to make any assumption about a Gaussian distribution, as the standard deviation here is just a descriptive statistic of the spread in the data that is defined (and non-zero) as long as we have at least two data points.

In the revised manuscript, we specified (in bold) that one point averages ~12 sondes: “For all terms, the sampling uncertainty is computed at the 3 h-scale as the standard error, $SE = \sigma/\sqrt{n}$, of the three individual 1 h circle values **(each representing ~50min of flight or up to 12 sondes)**, where σ is the standard deviation and n the number of circles.”

- L557: Is there also measurement uncertainty for each estimate of C that you add in quadrature? Or if not, why do you ignore this? No, we do not include measurement uncertainty of the individual C estimates, although there is uncertainty associated with them (e.g. regarding the choice of the extinction distance and reflectivity thresholds (in C and C_{only}), the particle size and liquid water content thresholds (in C_{pma}), or the relative humidity threshold (in C_{turb}), as noted in Bony et al. (2022)). But because it is difficult (and very subjective) to quantify these sensitivities, and because sensitivity tests suggest the individual measurement uncertainties are smaller compared to the differences among the different C estimates—which differ in terms of measurement technique and spatial sampling—, we decided to account only for the latter.

To clarify this issue, we modified the discussion of the C uncertainty in the Methods Sec. *Uncertainty estimation* of the revised manuscript:

“For C, the estimation uncertainty is computed for every 3h-circle set as the SE of the four different estimates of C, namely C itself, C_{only}, C_{turb}, and C_{pma}. The uncertainty estimate therefore

represents uncertainty in measurement principles and spatial sampling (Bony et al. 2022). **Additional uncertainties of the individual C estimates (e.g. due to the choice of thresholds) are neglected, as sensitivity tests suggest they are smaller than the uncertainty among the different C estimates (Bony et al. 2022)."**

• L590: Do you also assume here density is constant? Or if you have a measurement of density, why do you say you assume it constant in L509?

As we define M as the (mass) specific mass flux, with units of m/s, we also neglect contributions from density variations to variability in M, as is commonly done. This is justified by the small variations in the density at h (mean±sd of 1.104 ± 0.0077 kg/m³, i.e. less than 0.7% of the mean), which is due to the small variation of h and surface pressure across the measurements. To make this justification clearer, we rephrased the sentence in L509 to (new in bold, old as strikethrough):

"Also note that our M is defined as the (mass) specific mass flux and has units of velocity. It differs from the more familiar mass flux (in units of kg m⁻² s⁻¹) by the air density, which is usually assumed to be constant^{18,59}, and which is justified here given the small **density variations across the measurements (mean±sd of 1.104 ± 0.0077 kg/m³, i.e. less than 0.7% of the mean)**, ~~variation of h (Extended Data Fig. 3a) and surface pressure (less than 0.2% of the mean) across the measurements.~~"

• L620: 0.25% seems very small. What is the argument to include these profiles at all rather than impose a threshold for minimum cloudiness to consider from the GCMs? True, the threshold is very small, but the results are practically identical for e.g. a 1% threshold. We found 0.25% to be a good compromise, and used it as some models had very low cloud fractions. We didn't want to exclude those cases, but wanted to make sure that the cloud-base level remains robustly defined (as this is also the level where M and R are evaluated).

• L641: You use the same regime definitions for amip and amip4K, right? I assume you chose the grid points to compare based on if this trade Cu definition held in the base simulations (amip) and then look at the same grid points in amip4K. Please clarify. Given that we are considering the *thermodynamic* component of the feedback, we are computing the CRE of the trade cumulus regimes composited for the amip and amip4K climates separately. We thus apply the trade cumulus regime definition to each climate, not only the present-day climate. Otherwise the response could include a *dynamical* feedback component. The feedback is then defined as the difference between these two composites. We clarified this in the manuscript as follows:

"...35°S to 35°N, and use the regime partitioning of ref⁴⁰ with trade cumulus regimes in each simulation (amip or amip4K) defined as having a climatological annual mean estimated....".

Comments on figures/captions:

(Please note that by moving the schematic (ED Fig. 1) to the main text as Fig. 1, the numbers of the Figures in the second revised manuscript have changed compared to the first revision.)

• Fig 1.

o What are the "X" symbols in panel d? The green 'x' symbols indicate the data points that are excluded in the correlations due to inconsistent sampling of the mesoscale cloud patterns between the two aircraft. We added the following sentence to the legend: "The R in d is shown for both the HALO (blue) and ATR (green) aircraft, **with the x markers representing the data points that are excluded in the correlations due to inconsistent sampling of the mesoscale cloud patterns between the two aircraft.**"

o Do the relative humidity measurements also have observational uncertainty? Can that be added to panel d? The manufacturer (Vaisala) stated that the repeatability, i.e. the standard deviation of differences in twin soundings, of the RH estimates is 2%¹. Additional uncertainty

¹ <https://www.vaisala.com/sites/default/files/documents/RD41-Datasheet-B211706EN.pdf>

can come from the correction of the dry bias in the HALO dropsondes (see George et al. 2021). As we cannot distinguish uncertainty of the different data points, and because the manufacturer stated uncertainty is a default value unspecific to the EUREC⁴A campaign, we decided not to include error bars in the figure. However, we added the following statement about the RH uncertainty in the Methods Sec. *Uncertainty quantification*:

“For R, the manufacturer stated uncertainty (i.e., repeatability) is 2% and some additional uncertainty stems from the correction of the dry bias of the HALO dropsondes (see George et al. 2021). Because this uncertainty is the same for all data points, the estimation uncertainty of R is not shown in the figures.”

• Fig 2.

o Can error bars be added to panel a) on the reconstructed C values? We also wondered if we can add meaningful error bars to the reconstructed C values, like the standard error of the regression. But eventually we decided to leave them away and rather show the 1:1 line, as the distance between the data points and the 1:1 line (together with the correlation coefficient) already indicates the goodness of the regression fit.

o Why are there no error bars shown on panel b) when these data are already shown in c) and d)? We omitted the error bars in panel b) to make sure the figure is not overloaded (see below), and because the information is already shown in panels c) and d). The reader can thus refer to these panels to get an idea of the uncertainty.

o It would be nicer to have the r value in the top left of every panel, including b). As there is no space in the top left of panel b, we moved all r-values to the bottom left to make it consistent for all panels.

• Fig 3.

o It might be better to color the models by their feedback strength (not binned) and then show the color bar explicitly? We tried that, but did not find a satisfactory color bar to display it explicitly. Because the feedback strength is shown in panel c) and the reader can refer to it to get the exact value for every model, we decided to leave the color scale as is.

o Panel c) What do you mean by “frequent stratocumulus”? Can you be quantitative here. We added our definition of ‘frequent stratocumulus’ to the legend: “Open symbols refer to models with frequent stratocumulus (defined as having $R > 94\%$ more than 15% of the time, see Extended Data Fig. 8a)”.

o I’m not sure how useful it is to modify the extreme outlier models to fit onto your axes. Maybe it is better to just leave them off and note the values in the caption? Or maybe it is better to increase the axes to accommodate all models? The way you have it on panels a) and c) is hard to parse because naturally our eyes will try and find trends between the models which are misrepresented here. We were also thinking of leaving them off, but then the information about the mean \pm sd of C for IPSL6 and the radiative feedback strength of BCC is also not shown. Increasing the axes to accommodate all models (and also the full obs 95%-CI) is an option, but then the most meaningful content of the Figure gets lost and the models with the more reasonable values fall onto each other (see below).

So we decided to stick to modifying the extreme outliers, but to a slightly lesser degree than in the initial version (see new Figure below). This hopefully alleviates the issues a bit.

o Panel c) – It seems like the observational uncertainty bars are not shown on the plot for the high end of a_M/a_R . Please fix. True, panel c) didn't include the full 95%-CI because it would stretch the scale a lot (see Figure above). In the revised caption, we now note the upper end of the 95%-CI.

The new figure including caption is here:

Fig. 4 | Relationships in climate models and link to trade cumulus feedback. **a**, Mean $\pm \sigma/2$ of M and C , **b**, correlation coefficients r between M and C ($r_{M,C}$) and R and C ($r_{R,C}$), and **c**, ratio of the standardized multiple linear regression coefficients a_M/a_R and the thermodynamic component of the trade cumulus radiative feedback. The models are colored in bins of feedback strength. Open symbols refer to models with frequent stratocumulus (defined as having $R > 94\%$ more than 15% of the time, see Extended Data Fig. 8a). The grey shading represents the 25th to 75th quartile and the grey bars the 95%-CI of bootstrapped observational values. For plotting purposes, **a** shows the mean $\bar{M}-30$ for IPSL-CM6A, and **c** shows the ratio a_M/a_R+3 for BCC-CSM2. In **c**, the upper end of the observational 95%-CI (at 6.75) is cropped.

• ED Fig 1. I still really like this figure. Is there space to add it to the main text? **Yes, there is space! We moved it to the main text.**

• ED Fig 2.

o Panel a) should have M' not M in the legend. Check this in other figures too. **We changed this in all the figures.**

o Panel b) please update the terminology of “circling” vs “circle” to be consistent with the changes to “1h” and “3h” **Changed here and in all other figures**

o Can you change “r” and “r.c” to “r_{1h}” and “r_{3h}” for the correlation coefficients? We kept r for the default 3h-scale and changed r.c to r.1h for the 1h-scale. We also changed this in the other figures.

o It would be nice to show the error bars on these scatter plots as you have in Fig 2 We added the error bars and adjusted the Figure legend to include all the changes applied.

• ED Fig 3. What are the dashed lines on panels b) and d)? We added the explanation of the dashed lines to the Figure legend: “The dashed lines in **b** show the mean and standard deviation of E from ref²², and the zero line in **d**.”

• ED. Fig 4. Like Fig 1., what are the “X” symbols? As for Fig. 1, we added the explanation in the legend.

• ED. Fig 5.

o Change legend “circle and circling” Changed.

o What is “10m” on panel h)? Can you just write out “10-m WS” or something more descriptive? Changed to “10m wind”.

o What are the grey symbols here? Please restate this definition in the figure caption. The grey symbols represent the data that wasn’t used for comparing the relationships between M , R and C , either because of inconsistent sampling between the two aircraft (points marked with x in Fig. 2&3) or because ATR wasn’t flying at that time. As the E , W and R data in the panel comes entirely from HALO, however, we found it useful to also show the correlation of this larger sample. To make this clearer for the reader, we rephrased the definition in the Figure legend (see below).

o What are the “x” vs “o” markers indicating? Please add this to the caption. We removed the x markers to make the figure a bit less complicated.

o Note in caption what the dotted lines are in panels a) and g). 1:1, yes? Added. The new Figure legend is as follows:

Extended Data Fig. 4 | Relationships of other key terms. a, E and W versus M , **b**, surface buoyancy flux versus E , **c**, $\Delta\theta_v$ versus E , **d**, \mathcal{R} versus E , **e**, \mathcal{R} versus W , **f**, BVA mixing indicator versus M , **g**, E versus W , and **h**, 10 m wind speed versus surface buoyancy flux. **a-c,f-h** show both the 3 h-scale (filled) and 1 h-scale (open circles, with the corresponding correlation coefficient denoted as ‘r.1h’). A dotted 1:1 line is shown in **a** and **g**. In **d-e**, the error bars represent the estimation uncertainty for E and W , and the sampling uncertainty for \mathcal{R} (see Methods). The correlations in **d-e** are given both for the sample with consistent sampling among the HALO and ATR aircraft (blue points, as used for the correlations in Fig. 3), and for the entire sample of the HALO aircraft (including the grey points that represent the three data points marked with x in Fig. 3, and 8 other data points when ATR was not flying. The corresponding correlation coefficient is denoted as ‘r.all’).

• ED. Fig. 6.

o Correct M' to M Changed.

o Make caption text describing a) and b) parallel. We simplified the caption:

Extended Data Fig. 5 | Influence of different M and C estimates on key relationships. Correlation coefficients r of **a**, M and C ($r_{M,C}$) and M and \mathcal{R} ($r_{M,\mathcal{R}}$) and **b**, M and C ($r_{M,C}$) and \mathcal{R} and C ($r_{\mathcal{R},C}$). **c-d**, correlations of the reconstructed $\hat{C} = a_0 + a_M \tilde{M} + a_{\mathcal{R}} \tilde{\mathcal{R}}$ and the observed C ($r_{\hat{C},C}$), as well as the ratio of the standardized regression coefficients $a_M/a_{\mathcal{R}}$. **a** and **c** also show the relationships for the total M' (open symbols), whereas **b** and **d** show the relationships for different estimates of C (different symbols). See details in Methods subsection ‘Robustness of observational estimates’.

- ED. Fig. 7. Elsewhere cloud cover is written just as “C” not “CC” “CC” refers to the total projected cloud cover, while “C” refers to the cloud fraction at cloud base. We added in the Figure legend title: “Relationship of M with three estimates of the total projected cloud cover (CC).”
- ED. Fig. 9.
 - o Change 1:1 lines to dotted to match other figures and include this in the caption. Changed
 - o Indicate that the colors of the markers matches previous figures and they are colored by feedback strength. Changed
 - o Are “stratocumulus-like conditions” just when $R > 94\%$? Or is there another criteria? This is unclear to me from the text and the caption. Changed.
 - o “standard deviation σ of C”  “standard deviation of C (σ_C)” Changed. The new Figure legend is as follows:

Extended Data Fig. 8 | Comparison of other variables and relationships in climate models against the EUREC4A data. **a**, mean \mathcal{R} and fraction of *stratocumulus-like* conditions with $\mathcal{R} > 94\%$, **b**, standard deviation of \mathcal{R} and W ($\sigma_{\mathcal{R}}$ and σ_W), **c**, r^2 of multiple linear regression $\hat{C} = a_0 + a_M \tilde{M} + a_{\mathcal{R}} \tilde{\mathcal{R}}$ and correlation coefficient of M and \mathcal{R} , **d**, standard deviation of C (σ_C) and thermodynamic component of the cloud feedback $\Delta CRE / \Delta T_s$, as well as the 3 h and monthly correlations of **e**, M and C , and **f**, \mathcal{R} and C . **e-f** also show the inter-model correlation coefficients of the respective variables and the 1:1 line (dotted). As in Fig. 4, the models are colored in bins of feedback strength, and open symbols indicate models with frequent stratocumulus (defined as having $\mathcal{R} > 94\%$ more than 15% of the time). The observational uncertainty range is shown in grey, with the shading representing the 25th to 75th quartile and the grey bars the 95%-CI of bootstrapped values. HadGEM2-A is not shown in **b** due to the absence of W output.

References

- Bony, S. et al. Eurec4a observations from the safire atr42 aircraft. *Earth Syst. Sci. Data* **14**, 2021–2064 (2022)
- George, G. et al. Joanne: Joint dropsonde observations of the atmosphere in tropical north atlantic meso-scale environments. *Earth Syst. Sci. Data* **13**, 5253–5272 (2021)
- Myers, T. A. et al. Observational constraints on low cloud feedback reduce uncertainty of climate sensitivity. *Nature Climate Change* **11**, 501–507 (2021).
- Savazzi, A. C. M., Nuijens, L., Sandu, I., George, G. & Bechtold, P. The representation of winds in the lower troposphere in ecmwf forecasts and reanalyses during the eurec4a field campaign. *Atmos. Chem. Phys. Disc.* **2022**, 1–29 (2022).
- Stevens, B., Sherwood, S. C., Bony, S. & Webb, M. J. Prospects for narrowing bounds on earth’s equilibrium climate sensitivity. *Earth’s Future* **4**, 512–522 (2016)
- Vial, J., Vogel, R., & Schulz, H. On the daily cycle of mesoscale cloud organization in the winter trades. *Q. J. R. Meteorol. Soc.* **147**, 2850–2873 (2021)